

# Boundary vertex algebras for 3d $\mathcal{N}=4$ rank-0 SCFTs

Andrea E. V. Ferrari[1][⋆], Niklas Garner[2][†] and Heeyeon Kim[3][‡]

**1** School of Mathematics, James Clerk Maxwell Building, Mayfield Road,
University of Edinburgh, Edinburgh, EH9 3FD, United Kingdom
**2** Department of Physics, University of Washington, Seattle, WA, 98195, USA
**3** Department of Physics, Korea Advanced Institute of Science and Technology,
Daejeon 34141, Republic of Korea

⋆ andrea.e.v.ferrari@gmail.com , † nkgarner@uw.edu , ‡ heeyeon.kim@kaist.ac.kr

## Abstract

We initiate the study of boundary Vertex Operator Algebras (VOAs) of topologically twisted 3d $\mathcal{N}=4$ rank-0 SCFTs. This is a recently introduced class of $\mathcal{N}=4$ SCFTs that by definition have zero-dimensional Higgs and Coulomb branches. We briefly explain why it is reasonable to obtain rational VOAs at the boundary of their topological twists. When a rank-0 SCFT is realized as the IR fixed point of a $\mathcal{N}=2$ Lagrangian theory, we propose a technique for the explicit construction of its topological twists and boundary VOAs based on deformations of the holomorphic-topological twist of the $\mathcal{N}=2$ microscopic description. We apply this technique to the $B$ twist of a newly discovered family of 3d $\mathcal{N}=4$ rank-0 SCFTs $\mathcal{T}_r$ and argue that they admit the simple affine VOAs $L_r(\mathfrak{osp}(1|2))$ at their boundary. In the simplest case, this leads to a novel level-rank duality between $L_1(\mathfrak{osp}(1|2))$ and the minimal model $M(2,5)$. As an aside, we present a TQFT obtained by twisting a 3d $\mathcal{N}=2$ QFT that admits the $M(3,4)$ minimal model as a boundary VOA and briefly comment on the classical freeness of VOAs at the boundary of 3d TQFTs.

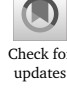

# 1 Introduction

The past few years have seen exciting progress in the study of the topological $A$ and $B$ twists of 3d $\mathcal{N}=4$ theories, see *e.g.* [1–9] for a small selection of results related to this work. The resulting non-unitary TQFTs exhibit in general more exotic features than the more familiar TQFTs of Schwarz type, such as Chern-Simons theory. For instance, their state spaces are not finite-dimensional [10, 11], and their categories of line operators are expected to form intricate, non-semisimple braided-tensor or $E_2$ categories [12–14], which are much less rigid than the Modular Tensor Categories (MTCs) that describe more standard TQFTs.[1] Moreover, much like their more standard topological cousins, topologically twisted 3d $\mathcal{N}=4$ theories enjoy holomorphic boundary conditions supporting vertex operator algebras (VOAs); the first explicit examples of these boundary VOAs appear in [17] and were expanded upon in [2], see also [15, 18] for detailed studies of abelian $\mathcal{N}=4$ gauge theories of hypers and vectors. Since the bulk line operators do not necessarily form a semisimple category, these VOAs are generically not rational, *i.e.* these theories admit logarithmic VOAs on their boundaries.

Much progress in the study of these rather exotic TQFTs has been made possible by exploiting algebro-geometric tools closely related to the geometry of their moduli spaces of vacua. Familiar examples where this has been fruitful are Rozansky-Witten theory [19], *cf.* [20,21], as well as the sharp mathematical definition of Coulomb branches due to Braverman-Finkelberg-Nakajima [22,23], *cf.* [24–26]. It is therefore natural to ask what features arise in the event that the Higgs and/or Coulomb branches of the 3d $\mathcal{N}=4$ theory trivialize in a suitable sense. When both branches are zero dimensional these theories are said to be rank-0. Partial expectations have been formulated in this context [5,27], where evidence was provided to support the claim that the boundary VOAs are closely related to rational CFTs.

One common problem obstructing the study of theories with zero-dimensional Higgs and Coulomb branches is that it is not straightforward (and perhaps even impossible) to engineer them starting from a microscopic Lagrangian description with manifest $\mathcal{N}=4$ supersymmetry. In [5,27], for example, these theories are realized by either starting from a UV $\mathcal{N}=2$ theory that enhances in the IR or by starting with UV $\mathcal{N}=4$ theory and then gauging a symmetry that emerges only in the IR. These IR SCFTs can be studied by utilizing protected observables,

---

[1]One particularly promising way to understand these categories in some instances is in terms of the representation theory of quantum groups, see *e.g.* [6,15,16].

such as superconformal indices, half-indices, and sphere partition functions, and tuning certain parameters in a way that corresponds to the would-be topological $A$ or $B$ twist.

The present paper reports the results of our first efforts to provide a more direct understanding of these strongly interacting rank-0 $\mathcal{N} = 4$ SCFTs, focusing in particular on their boundary VOAs. Of course, these VOAs are much more sensitive than standard supersymmetric observables and, in fact, much of the spectrum of the bulk theory can in principle be reconstructed from them. The basic idea that we utilize to directly access these topologically twisted theories and their boundary VOAs is analogous to the perspective taken in [6–8] for 3d theories (see also [28,29]) and [30,31] in 4d. We start by first passing to the holomorphic-topological ($HT$) twist of the UV Lagrangian theory, which is available to any $\mathcal{N} \geq 2$ theory, and identify a suitable boundary vertex algebra utilizing the tools provided by [32]. We then study how this is deformed upon passing to a fully topological theory.

The underlying intuition is that the $HT$ twist already captures IR information, and should therefore be able to see the emergent supersymmetry, *cf.* [33]. The existence of extra supercurrents extending $\mathcal{N} = 2$ to $\mathcal{N} = 4$, and therefore of two *topological* twists in the IR, manifests itself in the $HT$ twist as the presence of two distinguished operators whose descendants can be used to deform the holomorphic-topological theory to two theories that are fully topological. Roughly speaking, this deformation can be thought of as turning on a suitable superpotential. From a BV/BRST perspective, this deforms the differential $Q$ of the $HT$-twisted theory, which is the sum of the BV/BRST differential and the twisting supercharge $Q = Q_{BRST} + Q_{HT}$, to $Q_{A/B} = Q + \delta_{A/B}$. The effect of such a superpotential deformation on the boundary vertex algebra can be understood using the analysis of [32]. As we review below, see also [7,33], the second operator plays a distinguished role after deforming: it trivializes (makes homotopically trivial) the bulk's dependence on the holomorphic coordinate and oftentimes realizes an action of the Virasoro algebra on boundary local operators (thus producing a VOA). Importantly, the form of the superpotential used to deform to topological theories instructs which boundary conditions are deformable in the sense of [2,34].

In this paper, we systematically apply this strategy to one particularly simple but remarkable class of rank-0 SCFTs that has recently been announced in [27]. The simplest member of this family is the so-called minimal SCFT $\mathcal{T}_{\min}$ of [35] whose $A$ and $B$ twists were studied in [5]. This class corresponds to the IR image of a certain family of abelian $\mathcal{N} = 2$ Chern-Simons-matter theories. More precisely, we consider the theory $\mathcal{T}_r$ of the following form

$$\mathcal{T}_r \; : \; \mathcal{N} = 2 \quad U(1)^r_{K_r} + \Phi_{a=1,\cdots,r} \, , \qquad W = \sum_{i=1}^{r-1} V_{\mathfrak{m}_i} \, , \tag{1}$$

where we have $r$ copies of a $U(1)$ gauge theory with a chiral multiplet of charge 1, a suitable matrix $K_r$ of mixed Chern-Simons level, and $V_{\mathfrak{m}_i}$ are a basis of gauge-invariant half-BPS monopole operators. In the IR, a $U(1) \subset (U(1)^r)_{\text{top}}$ subgroup of the topological symmetry mixes with the $\mathcal{N} = 2$ R-symmetry $U(1)_R$ to form the $\mathcal{N} = 4$ R-symmetry group $SU(2)_H \times SU(2)_C$; the remaining topological symmetries are broken by the monopoles operators, in agreement with the rank-0 expectation. One of the results of [27] was that certain half indices of these theories, in the limits appropriately reproducing the $A$ and $B$ twists, appear to be vector-valued modular forms closely related to characters of the Virasoro minimal models $M(2, 2r + 3)$. As we now explain, our main motivation to study these boundary VOAs came from the observation of these modular phenomena as well as the appearance of Virasoro minimal models. We note that these minimal models also arise in the context of 4d Argyres-Douglas theories [36] and their reduction to 3d [37,38].

Before zooming in on the results of this paper, some general comments on the expected properties of the VOAs appearing at the boundary of twisted rank-0 theories SCFTs are in order.

First, we remark that the vacuum geometry is expected to control two important aspects of the boundary VOAs in the $A/B$ twist of an $\mathcal{N} = 4$ theory.

i) Based on examples [3], the Higgs branch conjecture of Beem-Rastelli in 4d, as well as free-field realizations in 3d by Beem and the first author [39], the Higgs/Coulomb branch chiral ring identified with (the reduced part of) Zhu's $C_2$ algebra $R_\mathcal{V}$ of the boundary VOA $\mathcal{V}$; in [39] it was conjectured that for a large class of A-twisted abelian gauge theories, the associated variety $\mathrm{Specm}(R_\mathcal{V})$ is indeed the Higgs branch.

ii) Additionally, the Coulomb/Higgs branch is supposed to correspond to the algebra of self-extensions of the vacuum module of the boundary algebra [2, 3].

Thus, based on these expectations, it is natural to strengthen a bit the rank-0 conditions, and study the case where:

1) $R_\mathcal{V}$ is finite-dimensional

2) there are no non-trivial self-extensions of the vacuum module

The first condition, which is equivalent to saying $\mathcal{V}$ is $C_2$-cofinite, already has important consequences. In fact, it is known that under some technical assumptions (that the VOA is finitely strongly generated and non-negatively graded, which usually holds) the $C_2$-cofiniteness condition (and as a consequence, the zero-dimensionality of the associated variety) is equivalent to the vertex algebra being lisse [40], which says that the singular support of $\mathcal{V}$ (as a module for itself) is zero-dimensional. This further implies that its characters indeed enjoy modular properties, see *e.g.* [41]. Therefore, taking seriously the first expectation means that rank-0 theories have a good chance of admitting lisse boundary VOAs, and in turn, of having half-indices that enjoy modular properties.

Exploring in detail the second condition is beyond the scope of this paper. However, in part in view of the findings of [5, 27] we are prompted to formulate the following question:

**Q**: Under which conditions is a lisse VOA with trivial self-extensions of the vacuum module in fact rational?

To put it differently, it has been a long-standing problem in the VOA literature to elucidate the relation between the lisse and rational conditions. The absence of self-extensions of the vacuum is certainly a necessary condition for a VOA to be rational, and it seems reasonable to expect that it is generically also sufficient. Leaving a precise formulation and proof of this statement to future work, we limit ourselves to observe that conditions 1) and 2), which are expected to be in general stronger than the rank-0 condition, are in fact necessary to obtain a rational VOA. We call this the *strongly* rank-0 condition.

We will now describe in more detail the contents and results of this paper. We start with a brief review of the class of theories under consideration in Section 2 and discuss the generalities of deforming an $HT$-twisted theory to a full topological theory in Section 3.

In Section 4 we study the minimal rank-0 theory $\mathcal{T}_1 \equiv \mathcal{T}_{\min}$. We start by examining the $HT$ twist of $\mathcal{T}_1$ and describe the boundary vertex algebra supported by a (right) Dirichlet boundary condition Dir analogous to the one proposed by [27]. We find that this boundary condition is deformable to the $B$ twist and performing this deformation leads to an algebra of $\mathfrak{osp}(1|2)$ currents at level 1, *i.e.* the simple affine VOA $L_1(\mathfrak{osp}(1|2))$. As expected, this simple affine VOA is rational (and therefore satisfies the strongly rank-0 conditions). The representation theory of this VOA has been extensively studied and we identify its modules with bulk Wilson lines by equating their characters with refined half-indices. We then describe the fusion rules of these Wilson lines and the action of the modular group on the torus state space $\mathcal{H}(\Sigma_{g=1})$, viewed

as the space of (super)characters of $L_1(\mathfrak{osp}(1|2))$, finding results compatible with the analyses of [5,27]. Furthermore, we remark on a tantalizing connection to the Virasoro minimal model $M(2,5)$ in terms of a novel level-rank-like duality[2] based on embeddings into free fermions, and argue that a suitable treatment of the algebra of local operators on a (left) dressed Neumann boundary condition Neu should realize $M(2,5)$ as a boundary VOA.

We consider the higher-level theories $\mathcal{T}_r$ in Section 5, where a direct description of the boundary VOA becomes somewhat more difficult. We argue that there is a Dirichlet boundary condition $\mathrm{Dir}^{(r)}$ that is compatible with the $B$-twist and provide evidence for the algebra of boundary local operators being identified with the simple affine VOA $L_r(\mathfrak{osp}(1|2))$. We finish Section 5 by comparing properties of the category of $L_r(\mathfrak{osp}(1|2))$ modules with the analysis of [27], finding compatible results. The form of our half-indices can be viewed as fermionic sum representations of $L_r(\mathfrak{osp}(1|2))$ characters, which may be of independent interest. Finally, in Appendix A we make use of a mechanism introduced in Section 4 to propose a twisted 3d $\mathcal{N} = 2$ TQFT and boundary condition thereof that realizes the minimal model $M(3,4)$.

We note that the simple affine VOAs $L_r(\mathfrak{osp}(1|2))$ for $r \in \mathbb{Z}_{>0}$ are an interesting family of VOAs that are all lisse [42, Thm 5.5], rational [43, Thm 7.1], and classically free [44, Cor 3.1]. The first two properties were mentioned above and are quite natural from the perspective that the bulk TQFT arises from twisting a rank-0 SCFT. The notion of classical freeness was first introduced in [45, 46] in the computation of chiral homology groups, see also [47, 48] for further developments and examples, but is quite rare and still not well understood physically or mathematically. There is evidence that the VOAs coming from 4d $\mathcal{N} = 2$ SCFTs must be classically free, *e.g.* the only classically free minimal models are the $M(2, 2r + 3)$ [45] and these are precisely the ones realized by Argyres-Douglas theories of type $(A_1, A_{2n})$ [36]. This does not seem to hold, or at least needs to be modified, for VOAs on the boundary of 3d TQFTs: while the theories $\mathcal{T}_r$ admit classically free boundary VOAs, the theory studied in Appendix A admits one that is *not* classically free. It would be interesting to understand what properties of the bulk QFT are realized by the notion of classical freeness of its boundary VOA(s), or if it is possible for a given 3d QFT to admit boundary VOAs of both types.

## 2  3d $\mathcal{N} = 4$ rank-zero theories

In [5,35], Gang *et al.* constructed a class of 3d $\mathcal{N} = 4$ SCFTs of rank zero, which are characterized by the property that both their Higgs branch and Coulomb branch are point-like. These theories in general do not have a Lagrangian description that preserves the full $\mathcal{N} = 4$ supersymmetry. However, they often have a Lagrangian description with manifest $\mathcal{N} = 2$ symmetry, from which one can calculate various supersymmetric observables to study the properties of the low-energy theories. There is non-trivial evidence that allows us to conjecture that these $\mathcal{N} = 2$ theories flow to SCFTs in the infrared, where the supersymmetry enhances to $\mathcal{N} = 4$.

Such a rank-zero $\mathcal{N} = 4$ SCFT admits two topological twists, called the $A$ and $B$ twists, and each yields a semisimple topological quantum field theory (TQFT). The modular data of these TQFTs and characters counting boundary local operators can be extracted by computing their partition functions on Seifert manifolds (see *e.g.* [49] and references therein) and half-indices [50–52], respectively.

---

[2]We thank Thomas Creutzig for explaining this duality to us and suggesting a possible connection to our construction.

## 2.1 The minimal $\mathcal{N} = 4$ rank-zero SCFT, $\mathcal{T}_{\min}$

The simplest example of a rank-zero SCFT, which is known as the minimal $\mathcal{N} = 4$ SCFT, $\mathcal{T}_{\min}$, can be constructed from the following UV $\mathcal{N} = 2$ Lagrangian Chern-Simons-matter theory [35]:

$$U(1)_{k=3/2} + \Phi \,, \tag{2}$$

where $\Phi = (\phi, \lambda)$ is an $\mathcal{N} = 2$ chiral multiplet of gauge charge $+1$. The UV description enjoys global symmetry $U(1)_R \times U(1)_{\text{top}}$, where the second factor is the topological symmetry. The theory has two gauge invariant dressed monopole operators:

$$\phi^2 V_{-1} \,, \quad \bar{\lambda} V_{+1} \,, \tag{3}$$

where $V_{\mathfrak{m}}$ is the bare monopole operator with the gauge flux $\mathfrak{m} \in \mathbb{Z}$. It was first argued in *loc. cit.* that these operators belong to the extra super-current multiplets, which provides strong evidence that supersymmetry enhances to $\mathcal{N} = 4$ in the infrared. At the fixed point, the global symmetry $U(1)_R \times U(1)_{\text{top}}$ is expected to enhance to the full R-symmetry group $(SU(2)_H \times SU(2)_C)/\mathbb{Z}_2$.

## 2.2 A class of rank-zero theories

In the recent paper [27], an interesting class of rank-zero theories that generalizes $\mathcal{T}_{\min}$ was constructed. These theories have the following $\mathcal{N} = 2$ Lagrangian descriptions in terms of the abelian Chern-Simons matter theories:

$$\mathcal{T}_r \; : \; \mathcal{N} = 2 \quad U(1)_{K_r}^r + \Phi_{a=1,\cdots,r} \,, \qquad W = \sum_{i=1}^{r-1} V_{\mathfrak{m}_i} \,, \tag{4}$$

where the gauge group is $U(1)^r$ with the effective mixed Chern-Simons levels $K_r$ given by

$$K_r = 2 \begin{pmatrix} 1 & 1 & 1 & \cdots & 1 & 1 \\ 1 & 2 & 2 & \cdots & 2 & 2 \\ 1 & 2 & 3 & \cdots & 3 & 3 \\ \vdots & \vdots & \vdots & \ddots & \vdots & \vdots \\ 1 & 2 & 3 & \cdots & r-1 & r-1 \\ 1 & 2 & 3 & \cdots & r-1 & r \end{pmatrix} \,. \tag{5}$$

This coincides with $2C(T_r)^{-1}$, where $C(T_r)$ is the Cartan matrix of the tadpole diagram $T_r$, which is obtained by folding the $A_{2r}$ Dynkin diagram in half. The charges of the $a$-th chiral multiplet $\Phi_a$ under the $b$-th gauge group factor is $\delta_{ab}$. Finally, the theory is deformed by the monopole superpotential $W$, where $V_{\mathfrak{m}_i}$ are the bare monopole operators with fluxes

$$
\begin{aligned}
\mathfrak{m}_1 &= (2, -1, 0, \dots 0), \\
\mathfrak{m}_2 &= (-1, 2, -1, 0, \dots 0), \\
&\cdots \\
\mathfrak{m}_{r-1} &= (0, \dots, -1, 2, -1),
\end{aligned} \tag{6}
$$

which are the first $r-1$ rows of $C(A_r)$. One can check that these monopoles form a basis of gauge invariant bare monopole operators in this theory.

After deforming by this monopole superpotential, the flavor symmetry of $\mathcal{T}_r$ is broken from $U(1)^r$ to $U(1)$, which we call $U(1)_A$. They are linear combinations of the $U(1)$ topological symmetry $M_a$ for each gauge group factor:

$$A = \sum_{a=1}^{r} a M_a \,, \tag{7}$$

which will be identified with the axial $U(1)$ R-symmetry in the enhanced $\mathcal{N} = 4$ algebra. Note that the simplest example, $\mathcal{T}_{r=1}$ corresponds to the minimal SCFT $\mathcal{T}_{\min}$.

As in the $r = 1$ case, there is strong evidence that $\mathcal{T}_r$ flows to an $\mathcal{N} = 4$ SCFT that is rank-zero, *cf.* [27]. For example, the low energy theory has the following 1/4-BPS gauge-invariant dressed monopole operators,

$$\bar{\lambda}_r V_{(\mathbf{0}_{r-2}, -1, 1)} \,, \qquad \phi_1^2 \phi_2^2 \cdots \phi_r^2 V_{(-1, \mathbf{0}_{r-1})} \,, \qquad \text{for } r > 1 \,, \tag{8}$$

which are expected to belong to the extra super-current multiplet. The manifest $U(1)_R \times U(1)_A$ symmetry of the UV gauge theory enhances to $(SU(2)_H \times SU(2)_C)/\mathbb{Z}_2$, which is the $R$-symmetry group of $\mathcal{N} = 4$ theories. The IR theory then admits the two topological twists, which are expected to produce two non-unitary semisimple TFTs $\mathcal{T}_r^A$ and $\mathcal{T}_r^B$.

## 3 Twisting $\mathcal{N} = 4$ theories

All of the examples in the previous section have one thing in common: the rank-0 $\mathcal{N} = 4$ SCFT lives in the IR of a certain $\mathcal{N} = 2$ Lagrangian theory. It is generally quite hard to directly access the topological twists of such an IR SCFT theory because the UV theory only has manifest $\mathcal{N} = 2$ supersymmetry. The manifest $\mathcal{N} = 2$ supersymmetry does admit a supersymmetric twist, although it is not fully topological: such a twist trivializes translations in one real direction (say, $\partial_t = \partial_3$) as well as a complex linear combination of translations in a transverse plane (say, $\partial_{\bar{z}} = \frac{1}{2}(\partial_1 + i\partial_2)$), resulting in a theory that behaves partially topological and partially holomorphic. Such a twist is called a holomorphic-topological ($HT$) twist to distinguish it from a fully topological twist.

If the $\mathcal{N} = 2$ theory actually had $\mathcal{N} = 4$ supersymmetry, the additional supersymmetries surviving the $HT$ twist could then be used to deform to either the $A$ or $B$ twist. This has proven to be a fruitful route to describing the topological twists of more familiar 3d $\mathcal{N} = 4$ theories, *cf.* [7, 28, 29, 34]; see also [30, 31] for related deformations in 4d. Thankfully, the $HT$ twisted theory is an RG invariant (up to quasi-isomorphism, *cf.* Section 3.7 of [32]) and hence should admit two topological deformations corresponding to the two topological twists of the IR $\mathcal{N} = 4$ SCFT. In this way, we can bypass the lack of $\mathcal{N} = 4$ supersymmetry in the UV theory by first passing to the $HT$-twist; see *e.g.* [6, 8] for examples of this approach. We note that there are some aspects of the physical theory that are harder to extract from this perspective, *e.g.* turning on a real mass or FI parameter, but it is particularly well-suited to extracting the boundary vertex algebras of interest.

### 3.1 The $HT$ twist of an $\mathcal{N} = 4$ theory

We start by presenting some general features of the $HT$ twist of an $\mathcal{N} = 4$ theory, following the conventions of [7]. The 3d $\mathcal{N} = 4$ supersymmetry algebra has 8 real spinor supercharges $Q_\alpha^{a\dot{a}}$, where $a, \dot{a}$ are $\mathrm{Spin}(4)_R \simeq SU(2)_H \times SU(2)_C$ R-symmetry spinor indices and $\alpha$ is a $SU(2)_E$ Lorentz spinor index, and the following anti-commutators

$$\{Q_\alpha^{a\dot{a}}, Q_\beta^{b\dot{b}}\} = \epsilon^{ab} \epsilon^{\dot{a}\dot{b}} (\sigma^\mu)_{\alpha\beta} P_\mu \,, \tag{9}$$

where $(\sigma^\mu)^\alpha{}_\beta$ are the usual Pauli matrices and $\epsilon$ is the 2-index Levi-Civita tensor, with the convention $\epsilon^{+-} = \epsilon^{\dot{+}\dot{-}} = 1$. We raise/lower spinor indices as $\chi^\alpha = \chi_\beta \epsilon^{\beta\alpha}$ and $\chi_\alpha = \epsilon_{\alpha\beta} \chi^\beta$, where our convention is $\epsilon_{+-} = 1$ so that $\epsilon^{\alpha\gamma} \epsilon_{\beta\gamma} = \delta^\alpha{}_\beta$. Up to equivalences, this supersymmetry algebra admits three types of twists, *cf.* [53, 54]: there is a single holomorphic-topological ($HT$) twist

$$Q_{HT} = Q_+^{+\dot{+}} \,, \tag{10}$$

and two topological twists

$$Q_A = \delta_{\dot{a}}^{\alpha} Q_{\alpha}^{a\dot{+}}, \qquad\qquad Q_B = \delta_{\dot{a}}^{\alpha} Q_{\alpha}^{+\dot{a}}. \qquad (11)$$

The supercharge $Q_A$ (resp $Q_B$) is invariant under diagonal $SU(2)_H \times SU(2)_E$ (resp. $SU(2)_C \times SU(2)_E$) rotations and we call the corresponding supersymmetric twist the $A$ twist (resp. $B$ twist) and this diagonal copy of $SU(2)$ spin group the $A$-twisted spin (resp. $B$-twisted spin); we note that this is sometimes called the $H$ twist (resp. $C$ twist) to indicate which factor of the $R$-symmetry is used in the twisted spin.

We can write the $\mathcal{N} = 4$ supersymmetry algebra as two commuting copies of the $\mathcal{N} = 2$ algebra by setting $Q^1 = Q^{--}, Q^2 = -Q^{-+}$ and $\overline{Q}^1 = Q^{++}, \overline{Q}^2 = Q^{+-}$. We will identify the $\mathcal{N} = 2$ supersymmetry generated by $Q = Q^1, \overline{Q} = \overline{Q}^1$ with that of the UV $\mathcal{N} = 2$ theory. The 3d $\mathcal{N} = 2$ supersymmetry algebra generated by $Q, \overline{Q}$ only admits a single twist (up to equivalence): the holomorphic-topological ($HT$) twist $Q_{HT} = \overline{Q}_+$. Our conventions are such that the $HT$ twist of an $\mathcal{N} = 4$ theory is the same as its $HT$ twist when viewed as an $\mathcal{N} = 2$ theory via $Q, \overline{Q}$. We will identify the $\mathcal{N} = 2$ $R$-symmetry with the diagonal torus $U(1)_R \hookrightarrow SU(2)_H \times SU(2)_C$ generated by $R = R_H + R_C$; the supercharge $Q_{HT}$ has $U(1)_R$ $R$-charge 1 and is a scalar under a combined $U(1)_E \times U(1)_R$ rotation generated by $J = \frac{1}{2}R - J_3$ that we call the $HT$-twisted spin.

Some portions of the full $\mathcal{N} = 4$ supersymmetry algebra remain after taking the $HT$ twist. A more detailed discussion of the symmetries present in the $HT$ twist of an $\mathcal{N} = 4$ theory is presented in the recent work of the second author [33]. For starters, all of the momenta $P_\mu$ commute with $Q_{HT}$ and become symmetries of the $HT$ twist; of course, $P_t$ and $P_{\bar{z}}$ are $Q_{HT}$-exact, so we are left with the symmetry generated by $P_z$. The supercharge $Q_{HT} = Q_+^{+\dot{+}}$ is invariant under the anti-diagonal torus $U(1)_S \overset{\triangle}{\hookrightarrow} SU(2)_H \times SU(2)_C$ generated by $S = R_H - R_C$ so the $HT$ twist has such a $U(1)$ symmetry. Moreover, the supercharges $\delta_A = -Q_-^2 = Q_-^{-+}$ and $\delta_B = \overline{Q}_-^2 = Q_-^{+-}$ deforming $Q_{HT}$ to $Q_A$ and $Q_B$ commute with $Q_{HT}$

$$\{Q_{HT}, \delta_A\} = 0 = \{Q_{HT}, \delta_B\}. \qquad (12)$$

The $HT$ twist should thus naturally admit two square-zero fermionic symmetries $\delta_A$, $\delta_B$ with vanishing $U(1)_R$ charge, $HT$-twisted spin $\frac{1}{2}$ and $U(1)_S$ charges $-1, 1$. Importantly, these fermionic symmetries satisfy

$$\{\delta_A, \delta_B\} = P_z. \qquad (13)$$

The key property of these symmetries is that if we use them to deform the theory, *i.e.* twist the theory by $Q_{A/B} = Q_{HT} + \delta_{A/B}$, the resulting theory is topological: whereas only $P_t, P_{\bar{z}}$ are exact in the $HT$ twist, $P_z$ becomes exact after either of these deformations

$$\{Q_A, \delta_B\} = P_z = \{Q_B, \delta_A\}. \qquad (14)$$

Note that $Q_A$ does not have homogeneous $HT$-twisted spin, but it is invariant under the $A$-twisted spin generator $J_A = J + \frac{1}{2}S = R_H - J_3$; similarly, $Q_B$ is invariant under the $B$-twisted spin generator $J_B = J - \frac{1}{2}S = R_C - J_3$.

## 3.2 Supercurrents and superpotentials

In the untwisted theory, the holomorphic momentum is realized via a surface integral of the energy-momentum tensor. We will assume the stress tensor is symmetric $T_{\mu\nu} = T_{\nu\mu}$. The same remains true in the $HT$-twisted theory, but it also admits a second description in terms of a secondary product realized via holomorphic-topological descent, *cf.* Section 2.2 of [32]. The notion of holomorphic-topological descent is a mild generalization of that introduced by

Witten [55]. If $\mathcal{O}$ is any local operator, we define an $n$-form-valued local operator $\mathcal{O}^{(n)}$, with $\mathcal{O} = \mathcal{O}^{(0)}$, satisfying the following holomorphic-topological descent equation for any $n > 0$:

$$Q_{HT}\mathcal{O}^{(n)} = (Q_{HT}\mathcal{O})^{(n)} + \mathrm{d}'(Q_{HT}\mathcal{O})^{(n-1)}, \qquad (15)$$

where $\mathrm{d}' = \mathrm{d}\overline{z}\partial_{\overline{z}} + \mathrm{d}t\partial_t$. There are only two such descendants in 3d and they take the form

$$\mathcal{O}^{(1)} = -\mathrm{d}\overline{z}(Q_{\overline{z}}\mathcal{O}) - \mathrm{d}t(Q_t\mathcal{O}), \qquad \mathcal{O}^{(2)} = -\mathrm{d}\overline{z}\mathrm{d}t(Q_{\overline{z}}Q_t\mathcal{O}). \qquad (16)$$

where $Q_{\overline{z}} = \frac{1}{2}Q_+$ and $Q_t = -Q_-$, with the convention that the 1-forms $\mathrm{d}\overline{z}$ and $\mathrm{d}t$ are fermionic.

When $\mathcal{O}$ is $Q_{HT}$-closed, integrals of its descendants on closed submanifolds of spacetime give a natural class of higher-dimensional operators which, in turn, can be used to define secondary products on in the twisted theory [56]. For example, for any $Q_{HT}$-closed local operator $\mathcal{O}_1$ we define a new local operator by the formula

$$\{\{\mathcal{O}_1, \mathcal{O}_2\}\}(w, \overline{w}, s) = \oint_{S^2} \mathrm{d}z\,\mathcal{O}_1^{(1)}(z, \overline{z}, t)\mathcal{O}_2(w, \overline{w}, s), \qquad (17)$$

where the integration cycle is over a small $S^2$ surrounding the point $(w, \overline{w}, s)$. When $\mathcal{O}_2$ is also $Q_{HT}$-closed, the resulting local operator is $Q_{HT}$-closed by Stokes' theorem. This operation decreases $R$-charge by 1 and $HT$-twisted spin by 1, i.e. $\{\{\mathcal{O}_1, \mathcal{O}_2\}\}$ has $R$-charge $r_1 + r_2 - 1$ and $HT$-twisted spin $j_1 + j_2 - 1$. In fact, we can replace $\mathrm{d}z$ by $\mathrm{d}ze^{\lambda z}$ to get an operator $\{\{\mathcal{O}_{1\lambda}\mathcal{O}_2\}\}(w)$; as shown in [57], this gives local operators in the $HT$-twist a 1-shifted (or degree $-1$) $\lambda$-bracket. Equivalently, we could consider the tower of brackets

$$\{\{\mathcal{O}_1, \mathcal{O}_2\}\}^{(n)}(w, \overline{w}, s) = \oint_{S^2} \mathrm{d}z\,z^n\mathcal{O}_1^{(1)}(z, \overline{z}, t)\mathcal{O}_2(w, \overline{w}, s) \qquad (18)$$

for all $n \geq 0$; this bracket decreases $R$-charge by 1 and decreases $HT$-twisted spin by $n + 1$.

Let us return to the holomorphic translation mentioned earlier. In an $\mathcal{N} = 2$ theory, there are additionally supercurrents $G_{\alpha\mu}$ and $\overline{G}_{\alpha\mu}$; the $Q_{HT}$ variation of $G_{\alpha\mu}$ trivializes two components of $T_{z\mu}$

$$Q_{HT}G_{+z} = -2iT_{z\overline{z}}, \qquad Q_{HT}G_{-z} = iT_{zt}, \qquad (19)$$

so that we have

$$P_z = \oint \star(T_{z\mu}\mathrm{d}x^\mu) = -i\oint T_{zz}\mathrm{d}z\mathrm{d}t + Q_{HT}(...). \qquad (20)$$

The operator $T_{zz}$ is not $Q_{HT}$ closed. Instead, it is the first descendant of the $Q_{HT}$-closed operator $G = -\frac{i}{2}\overline{G}_{-z}$ (up to $Q_{HT}$-exact terms):

$$\mathrm{d}zG^{(1)} = -iT_{zz}\mathrm{d}z\mathrm{d}t + Q_{HT}(...), \qquad (21)$$

and, in particular, the $\partial_z$ derivative of any $Q_{HT}$-closed operator $\mathcal{O}$ can be realized as a secondary product with $G$:

$$\partial_z\mathcal{O} = \{\{G, \mathcal{O}\}\}, \qquad (22)$$

cf. Eq. (2.16) of [32]. More generally, the higher brackets with $G$ extend this to an action of all holomorphic vector fields

$$\mathcal{L}_{z^n\partial_z}\mathcal{O} = \{\{G, \mathcal{O}\}\}^{(n)}, \qquad (23)$$

where $\mathcal{L}_{z^n\partial_z}$ denotes the Lie derivative. The operator $G$ has $R$-charge 1 and $HT$-twisted spin 2, compatible with $z^n\partial_z$ having $R$-charge 0 and spin $1 - n$, and is called the higher stress tensor. In much the same way, the residual $\mathcal{N} = 4$ supersymmetries $\delta_{A/B}$ and the generator of $U(1)_S$ can be realized as the descent bracket with $Q_{HT}$-closed operators $\Theta_{A/B}$ and $\mathcal{S}$:

$$\delta_{A/B}\mathcal{O} = \{\{\Theta_{A/B}, \mathcal{O}\}\}, \qquad S \cdot \mathcal{O} = \{\{\mathcal{S}, \mathcal{O}\}\}. \qquad (24)$$

Taking into account the quantum numbers of $\delta_{A/B}$, we find that the (bosonic) operators $\Theta_{A/B}$ have $R$-charge 1 and twisted spin $\frac{3}{2}$, as well as $U(1)_S$ charge $\mp 1$. Including the higher brackets, the operators $G$, $\mathcal{S}$, $\Theta_{A/B}$ realize an action of the positive part of the 2d $\mathcal{N} = 2$ superconformal algebra [33].

Given a $Q_{HT}$-closed local operator $\mathcal{O}$, we can consider deforming the action by a "superpotential" term $\int d z \mathcal{O}^{(2)}$; when $\mathcal{O}$ is a holomorphic function $W$ of the chiral fields, this precisely reproduces the usual notion of a superpotential deformation. For this to preserve twisted spin and the $U(1)_R$-symmetry, we must require that the local operator $\mathcal{O}$ has $R$-charge 2 and twisted spin 1. The result of this deformation is to shift the action of $Q_{HT}$ by the secondary product with $\mathcal{O}$:

$$Q_{\mathcal{O}} = Q_{HT} + \{\{\mathcal{O}, -\}\}. \tag{25}$$

We now see how to deform the $Q_{HT}$ twist of an $\mathcal{N} = 4$ theory to the topological $Q_{A/B}$ twist. First, we redefine the twisted spin and the $R$-charge to be those of the topological $A/B$ twist. This ensures the operator $\Theta_{A/B}$ has the necessary quantum numbers; for example, in the $A$-twist we take the $R$-charge generated by $R - S = 2R_C$ and the twisted spin generated by $J + \frac{1}{2}S = R_H - J_3$. This change in twisted spin is accounted for in a modification of the higher stress tensor $G \to G_{A/B} = G \pm \frac{1}{2}\partial_{\bar{z}}\mathcal{S}$. Second, we deform the action by the superpotential term:

$$\int d z \, \Theta_{A/B}^{(1)}, \tag{26}$$

which implements the deformation of the differential.

## 3.3 Deformable boundary conditions

The final notion we want to review before moving to boundary vertex algebras is the notion of a deformable boundary condition formalized in [2]; see [1,17] for earlier examples. In brief, their aim was to study a half-BPS boundary condition preserving 2d $\mathcal{N} = (0,4)$ supersymmetry. As the topological supercharges are the sum of two supercharges with opposite 2d chirality, we see that such a $\mathcal{N} = (0,4)$ boundary condition cannot be compatible with either the $A$ or $B$ twist. Instead, a $(0,4)$ boundary condition is called deformable if it can be deformed, *e.g.* by adding the integral of a boundary local operator to the boundary action, to a one compatible with $Q_A$ or $Q_B$.

Without direct access to the supercurrents giving rise to $Q_A$, $Q_B$, it is hard to ask that a boundary condition be deformable at the level of the UV $\mathcal{N} = 2$ field theory. Thankfully, the notion of a deformable boundary condition can be reformulated in the $HT$-twisted theory [34]. The authors of *loc. cit.* show that an $HT$-twisted boundary condition is compatible with the deformation to the $A/B$ twist so long as the "superpotential" $\Theta_{A/B}$ vanishes on the boundary. From this perspective, the vanishing of the superpotential is required for the boundary condition to preserve supersymmetry.

A further requirement we will impose is that the other operator $\Theta_{B/A}$ is unconstrained on the boundary. This operator necessarily cannot be $Q_{A/B}$-closed. Instead, it trivializes the higher stress $G_{A/B}$:

$$Q_{A/B}\Theta_{B/A} = G_{A/B}. \tag{27}$$

This relation implies that if $\mathcal{O}(w, \overline{w})$ is any $Q_{A/B}$-closed boundary local operator, then

$$\oint_{HS^2} d z z^n G_{A/B}^{(1)}(z\bar{z}, t)\mathcal{O}(w, \overline{w}) - \oint_{\partial HS^2} d z z^n \Theta_{B/A}|_{(z, \bar{z})}\mathcal{O}(w, \overline{w}) \tag{28}$$

is another $Q_{A/B}$-closed local operator for any $n \geq 0$ due to Stokes' theorem. We expect this to extend the action of the holomorphic vector field $z^n\partial_z$ to the boundary. More generally,

we expect that the operator $\Theta_{B/A}|$ generates a Virasoro subalgebra of the boundary algebra of local operators. As we will see below, it does not necessarily agree with the boundary stress tensor, but it will be a crucial part thereof.

## 4  A boundary vertex algebra for $\mathcal{T}_{\mathbf{min}}$

We now return to the theory $\mathcal{T}_1 = \mathcal{T}_{\min}$. Based on the analysis of [5,35], we are lead to the expectation that the $Q_{HT}$-closed operators

$$\Theta_A = \overline{\lambda}_- V_{+1}, \qquad \Theta_B = \phi^2 V_{-1} \tag{29}$$

are the operators realizing the deformations to the $A$ and $B$ twist; the charge for the $U(1)_S$ symmetry is identified with (the negative of the) monopole number. In view of the analysis in [27], there is a natural candidate deformable boundary condition: we expect that the boundary condition imposing

- $(0,2)$ Dirichlet boundary conditions $\mathcal{D}$ on the Chern-Simons vector multiplet

- $(0,2)$ Dirichlet boundary conditions $D$ on the chiral multiplet

is deformable to the topological $B$-twist; we call this boundary condition $\mathrm{Dir} = (\mathcal{D}, D)$. Indeed, Dirichlet boundary conditions for the chiral multiplet impose $\phi| = 0$ and leave $\overline{\lambda}_-|$ unconstrained, therefore $\Theta_B| = 0$. Moreover, the Dirichlet boundary conditions on the vector multiplet further imply the bulk local operator $\Theta_A$ is unconstrained on Dir. The local operators on Dir are counted by the half-index [50–52], which is defined by

$$\mathbb{I}(q) = \mathrm{tr}_{\mathrm{Ops}}(-1)^{R_\nu} q^{\frac{R_\nu}{2}+J_3} x^T, \tag{30}$$

where Ops is the vector space of local operators on Dir, $T$ schematically denotes the generators of (a torus of) the boundary flavor symmetry with $x$ the corresponding fugacities, and

$$R_\nu = R - \nu S. \tag{31}$$

Here $R$ is the superconformal R-symmetry $R = R_H + R_C$ and $S = R_H - R_C$. In the present setting, we can identify $S = -A$ for $A$ the generator of the topological flavor symmetry, *cf.* Eqs. (8) and (9) of [27].

The boundary condition Dir has two flavor symmetries: the topological flavor symmetry $U(1)_A$ of the bulk theory (with corresponding fugacity $\eta$) and the boundary $U(1)_\partial$ flavor symmetry commensurate with a Dirichlet boundary condition (with corresponding fugacity $y$). The anomaly polynomial characterizing the effective Chern-Simons levels between these symmetries and the $R$-symmetry is

$$2\mathbf{f}^2 + 2(\mathbf{f}_{\mathrm{top}} - \mathbf{r})\mathbf{f}, \tag{32}$$

where $\mathbf{f}$ is the gauge field strength, $\mathbf{f}_{\mathrm{top}}$ is the field strength coupling to the topological flavor symmetry generated by $A$, and $\mathbf{r}$ is the field strength coupling to the $R$-symmetry. The half-index for $\mathcal{T}_{\min}$ with the boundary condition Dir is then

$$\mathbb{I}_{\mathrm{Dir}}(q; y, \eta, \nu) = \sum_{\mathbf{m} \in \mathbb{Z}} \frac{q^{\mathbf{m}^2} y^{2\mathbf{m}}[(-q^{1/2})^{\nu-1}\eta]^{\mathbf{m}}(y^{-1}q^{1-\mathbf{m}}; q)_\infty}{(q; q)_\infty}. \tag{33}$$

In this expression, the unusual factor of $(-q^{\frac{1}{2}})^{\mathbf{m}(\nu-1)}$ comes from two sources: the factor $(-q^{\frac{1}{2}})^{\mathbf{m}\nu}$ comes from the appearance of the topological symmetry in $R_\nu$, whereas the factor $(-q^{\frac{1}{2}})^{-\mathbf{m}}$ comes from the above mixed gauge–$R$-symmetry (effective) Chern-Simons term, which induces $R$-charge on operators with gauge magnetic charge.

At the level of the half-index, deforming to the $B$-twist amounts to sending $\eta \to 1$ and $v \to 1$, leading to

$$\mathbb{I}_B(q;y) = \mathbb{I}_{\text{Dir}}(q;y,1,1) = \sum_{\mathfrak{m} \in \mathbb{Z}} \frac{q^{\mathfrak{m}^2} y^{2\mathfrak{m}} (y^{-1} q^{1-\mathfrak{m}};q)_\infty}{(q;q)_\infty} \,, \tag{34}$$

which is expected to reproduce the vacuum character of the B-twisted boundary algebra. Note that in the limit $y \to 1$, the half-index reduces to the character of a non-vacuum module of the Virasoro minimal model $M(2,5)$,

$$\mathbb{I}_B(q;1) = \sum_{n \geq 0} \frac{q^{n^2}}{(q)_n} = \chi_{\alpha=1}^{M(2,5)}(q) \,. \tag{35}$$

where $n = -\mathfrak{m}$; the terms with $\mathfrak{m} > 0$ vanish due to the $q$-Pochhammer symbol $(q^{1-\mathfrak{m}};q)_\infty$ in the numerator.

The character of other modules can be obtained by inserting line operators. If we consider the Wilson line of charge $-1$ (together with the background Wilson loop for the $R$-symmetry that introduces an overall sign), we obtain

$$\mathbb{I}_B(q;y)[\mathcal{W}_{-1}] = -\sum_{\mathfrak{m} \in \mathbb{Z}} \frac{q^{\mathfrak{m}^2-\mathfrak{m}} y^{2\mathfrak{m}-1} (y^{-1} q^{1-\mathfrak{m}};q)_\infty}{(q;q)_\infty} \,. \tag{36}$$

This expression in the $y \to 1$ limit is proportional to the vacuum character of the Virasoro minimal model, *i.e.*,

$$\mathbb{I}_B(q;1)[\mathcal{W}_{-1}] = -\sum_{n \geq 0} \frac{q^{n^2+n}}{(q)_n} = -\chi_{\alpha=0}^{M(2,5)}(q) \,. \tag{37}$$

As we discuss momentarily, the two characters (34) and (37) transform as a vector-valued modular form, once we multiply appropriate modular anomaly prefactors.

Based on the discussion in Section 5, the generic Dirichlet boundary condition $(\mathcal{D}, D_c)$ considered in [27] is similarly deformable for the $A$ twist of $\mathcal{T}_{\min}$ – although $\Theta_A$ survives on Dir, for generic Dirichlet boundary conditions $D_c$ for chiral multiplet, which simply set $\phi| = c$, there is an additional differential that makes $\Theta_A$ trivial in cohomology. Moreover, the operator $\Theta_B$ is non-vanishing on this generic Dirichlet boundary condition, whereas it vanishes on Dir. We leave a deeper treatment of the algebra of local operators on the generic Dirichlet boundary condition of [27] for future work.[3]

---

[3]Another candidate for a deformable boundary condition for the $A$ twist is $(\mathcal{D}, N)$, imposing $(0,2)$ Dirichlet boundary conditions on the vector multiplet and $(0,2)$ Neumann boundary conditions for the chiral multiplet. The Neumann boundary conditions on the chiral multiplet impose $\bar{\lambda}_-| = 0$ which implies $\Theta_A = 0$. However, the boundary theory has anomalous $U(1)_H$, which is incompatible with the $A$-twist. This can be remedied by adding a boundary fermion with $R$-charge 1 and $U(1)_\partial$ charge $-1$.

The $HT$-twisted half-index for $(\mathcal{D}, N)$ takes the form

$$\mathbb{I}_{(\mathcal{D},N)}(q;y,\eta,v) = \sum_{\mathfrak{m} \in \mathbb{Z}} \frac{q^{\frac{1}{2}\mathfrak{m}^2} y^{\mathfrak{m}} [(-q^{\frac{1}{2}})^v \eta]^{\mathfrak{m}}}{(q;q)_\infty (y q^{\mathfrak{m}};q)_\infty}$$
$$= \left( \sum_{\mathfrak{m} \in \mathbb{Z}} \frac{(-1)^{\mathfrak{m}} q^{\frac{1}{2}\mathfrak{m}(\mathfrak{m}+v)} y^{\mathfrak{m}} \eta^{\mathfrak{m}}}{(q;q)_\infty} \right) \left( \sum_{n \geq 0} \frac{(-1)^n q^{-\frac{1}{2}n(n+v)} \eta^{-n}}{(q;q)_n} \right).$$

Deforming to the $A$-twist amounts to sending $\eta \to 1$ and $v \to -1$, leading to

$$\mathbb{I}_A(q;y) = \mathbb{I}_{(\mathcal{D},N)}(q;y,1,-1) = \left( \sum_{\mathfrak{m} \in \mathbb{Z}} \frac{(-1)^{\mathfrak{m}} q^{\frac{1}{2}\mathfrak{m}(\mathfrak{m}-1)} y^{\mathfrak{m}}}{(q;q)_\infty} \right) \left( \sum_{n \geq 0} \frac{(-1)^n q^{\frac{1}{2}n(1-n)}}{(q;q)_n} \right),$$

## 4.1 The $B$-twist of Dir

We now derive the algebra of boundary local operators on the boundary condition Dir after deforming to the $B$ twist. The category of modules for the VOA that we find will be our model for line operators in the $B$ twist of $\mathcal{T}_{\min}$. We start by reviewing the boundary vertex algebra in the $HT$ twist, including non-perturbative corrections by boundary monopole operators. We then describe the deformation to the topological $B$ twist and outline the relationship between the resulting VOA and the non-unitary minimal models described above.

### 4.1.1 The $HT$-twisted boundary vertex algebra

It is straightforward to derive the boundary vertex algebra on the above Dirichlet boundary condition, in large part because we are working with an abelian gauge theory. See [32] for general aspects of the boundary vertex algebras. First consider the perturbative local operators, *cf.* Section 7 of *loc. cit.*; these are generated by an abelian current $B$ and a fermionic field $\overline{\lambda} = \overline{\lambda}_-|$ of spin 1:

$$B(z)B(w) \sim \frac{2}{(z-w)^2} \,, \qquad B(z)\overline{\lambda}(w) \sim \frac{-\overline{\lambda}(w)}{z-w} \,, \qquad \overline{\lambda}(z)\overline{\lambda}(w) \sim 0 \,. \qquad (38)$$

The OPE of the abelian current $B$ with itself reflects the effective Chern-Simons $k_{eff} = 2$, the OPE between the current $B$ and the fermion $\overline{\lambda}$ encodes the fact that it has gauge charge $-1$, and the OPE of the fermion $\overline{\lambda}$ with itself is regular due to the absence of a bulk superpotential, *cf.* Eq. (5.28) of [32]. Each of these operators has spin 1.

The non-perturbative corrections to this perturbative answer come in the form of boundary monopoles. As a module for the perturbative algebra, these boundary monopoles can be identified with spectral flow modules; see *e.g.* [15,16,18,58] for a sampling of how spectral flow arises in the context of Dirichlet boundary conditions for 3d abelian gauge theories. There are spectral flow morphisms acting on the above operators as follows:

$$\sigma_{\mathfrak{m}}(B(z)) = B(z) - \frac{2\mathfrak{m}}{z} \,, \qquad \sigma_{\mathfrak{m}}(\overline{\lambda}(z)) = z^{-\mathfrak{m}}\overline{\lambda}(z) \,. \qquad (39)$$

At the level of modes, this translates to the following automorphism

$$\sigma_{\mathfrak{m}}(B_n) = B_n - 2\mathfrak{m}\delta_{n,0} \,, \qquad \sigma_{\mathfrak{m}}(\overline{\lambda}_n) = \overline{\lambda}_{n+\mathfrak{m}} \,, \qquad (40)$$

where $B = \sum B_n z^{-n-1}$ and $\overline{\lambda} = \sum \overline{\lambda}_n z^{-n-1}$. Note that $\sigma_{\mathfrak{m}_1} \circ \sigma_{\mathfrak{m}_2} = \sigma_{\mathfrak{m}_1+\mathfrak{m}_2}$ and, in particular, $\sigma_{\mathfrak{m}}^{-1} = \sigma_{-\mathfrak{m}}$.

Given a module $M$ of the perturbative algebra, we can construct a family of modules $\sigma_{\mathfrak{m}}(M)$ as follows: the underlying vector spaces are isomorphic and for any module element $|\varphi\rangle \in M$ we denote the corresponding element of $\sigma_{\mathfrak{m}}(M)$ by $\sigma_{\mathfrak{m}}(|\varphi\rangle)$; the module structure is then defined by the formula

$$\mathcal{O}\sigma_{\mathfrak{m}}(|\varphi\rangle) = \sigma_{\mathfrak{m}}(\sigma_{-\mathfrak{m}}(\mathcal{O})|\varphi\rangle) \,. \qquad (41)$$

Of particular importance for us are the spectral flows of the vacuum module for the perturbative algebra. If $|0\rangle$ is the vacuum vector, satisfying $B_n|0\rangle = \overline{\lambda}_n|0\rangle = 0$ for $n \geq 0$, then we

---

which has unbounded powers of $q$. If we add a boundary Fermi multiplet with $U(1)_\partial$ and $U(1)_R$ charges $-1$ and $1$ respectively, as required to cancel the boundary anomaly, we need to multiply mth summand in the half-index by

$$(yq^{\mathfrak{m}};q)_\infty (q^{1-\mathfrak{m}}y^{-1};q)_\infty \,.$$

This has the pleasing effect of making the $q$-expansion bounded from below.

find

$$
B_n \sigma_{\mathfrak{m}}(|0\rangle) = \begin{cases} 0 & n > 0 \\ 2\mathfrak{m} & n = 0 \\ \sigma_{\mathfrak{m}}(B_n|0\rangle) & n < 0 \end{cases}, \qquad \overline{\lambda}_n \sigma_{\mathfrak{m}}(|0\rangle) = \begin{cases} 0 & n - \mathfrak{m} \geq 0 \\ \sigma_{\mathfrak{m}}(\overline{\lambda}_{n-\mathfrak{m}}|0\rangle) & n - \mathfrak{m} < 0 \end{cases}. \quad (42)
$$

Denoting the operators corresponding to the states $\sigma_{\mathfrak{m}}(|0\rangle)$ by $\nu_{\mathfrak{m}}(z)$, these expressions translate to the following OPEs:

$$
B(z)\nu_{\mathfrak{m}}(w) \sim \frac{2\mathfrak{m}\nu_{\mathfrak{m}}(w)}{z-w}, \qquad \overline{\lambda}(z)\nu_{\mathfrak{m}}(w) \sim (z-w)^{\mathfrak{m}}\sigma_{\mathfrak{m}}(\lambda_{-1}|0\rangle)(w) + \dots \quad (43)
$$

The OPEs of the boundary monopoles $\nu_{\mathfrak{m}}$'s with one another are simply those of ordinary vertex operators:

$$
\nu_{\mathfrak{m}_1}(z)\nu_{\mathfrak{m}_2}(w) = (z-w)^{2\mathfrak{m}_1\mathfrak{m}_2}:\nu_{\mathfrak{m}_1}(z)\nu_{\mathfrak{m}_2}(w): \sim (z-w)^{2\mathfrak{m}_1\mathfrak{m}_2}\nu_{\mathfrak{m}_1+\mathfrak{m}_2}(w) + \dots \quad (44)
$$

We note that the subalgebra generated by $B$ and the boundary monopoles $\nu_{\mathfrak{m}}$ can be identified with a rank-1 lattice VOA at level 2, *i.e.* based on the lattice $\sqrt{2}\mathbb{Z}$. This particularly important lattice VOA can be identified with the symmetry algebra of a compact boson at its critical radius: (the simple quotient of) an affine $\mathfrak{sl}(2)$ current algebra at level 1. In particular, if we identify $B \leftrightarrow h$, $e \leftrightarrow V_1$, and $f \leftrightarrow V_{-1}$, Eq. (44) translates to the following OPEs:

$$
\begin{aligned}
h(z)h(w) &\sim \frac{2}{(z-w)^2}, & e(z)f(w) &\sim \frac{1}{(z-w)^2} + \frac{h(w)}{z-w}, \\
h(z)e(w) &\sim \frac{2e(w)}{z-w}, & h(z)f(w) &\sim \frac{-2f(w)}{z-w}.
\end{aligned} \quad (45)
$$

We note that the spins of these operators are not the natural ones where each gets spin 1: we take $e$ to have spin $1 + \frac{1}{2}(\nu - 1)$ and $f$ spin $1 - \frac{1}{2}(\nu - 1)$, *cf.* Eq. (33).

Equation (43) implies that that the OPE of $f$ and $\overline{\lambda}$ is non-singular, whence it is a lowest weight vector of a doublet for this affine $\mathfrak{sl}(2)$ symmetry; the state $\overline{\lambda}_0 \sigma_1(|0\rangle) = \sigma_1(\overline{\lambda}_{-1}|0\rangle)$ fills out the remaining states in this doublet and we identify it with the restriction of the bare bulk monopole $V_1|$. (The bare bulk monopole $V_{-1}|$ is identified with $f$.) If we denote the operator dual to this state $\theta_+$ and rename $\overline{\lambda} = \theta_-$, Eq. (43) takes the form

$$
\theta_-(z)e(w) \sim \frac{\theta_+(w)}{z-w} \rightsquigarrow e(z)\theta_-(w) \sim \frac{-\theta_+(w)}{z-w}. \quad (46)
$$

The remaining OPEs of the fermionic and bosonic generators ultimately follow from the affine $\mathfrak{sl}(2)$ symmetry. More explicitly, the OPEs of $\theta_{\pm}$ and $h$ follow from Eq. (38) and (43):

$$
h(z)\theta_{\pm}(w) \sim \frac{\pm\theta_{\pm}(w)}{z-w}. \quad (47)
$$

Using the fact that $|\theta_+\rangle = -e_0|\theta_-\rangle$ together with $f_n|\theta_-\rangle = 0$ and $h_n|\theta_-\rangle = -\delta_{n,0}|\theta_-\rangle$ for all $n \geq 0$, we conclude

$$
f_n|\theta_+\rangle = h_n|\theta_-\rangle = -\delta_{n,0}|\theta_-\rangle \rightsquigarrow f(z)\theta_+(w) \sim \frac{-\theta_-(w)}{z-w}. \quad (48)
$$

That the OPE of $e$ and $\theta_+$ is non-singular follows from the fact that $e_n|\theta_-\rangle = 0$ for all $n > 0$ and that $e_0^2|\theta_-\rangle = 0$ because there are no states with the necessary charge an spin. Finally, we note that the OPE of $\theta_+$ and $\theta_-$ is non-singular; this follows from the action of $\overline{\lambda}_n$ on $\sigma_1(\overline{\lambda}_{-1}|0\rangle)$.

We expect that the algebra of boundary local operators is (strongly, but not freely) generated by the bosonic operators $e, f, h$ and the fermionic operators $\theta_\pm$.

We note that the spin $2 + \frac{1}{2}(\nu - 1)$ operator $\frac{1}{2}\epsilon^{\beta\alpha} : \theta_\alpha \theta_\beta :=: \theta_2 \theta_1 :$ of magnetic charge 1 has non-singular OPEs with all other fields. As explained in [32], operators in the center of a boundary vertex algebra arise from the restriction of bulk local operators to the boundary: these bulk local operators necessarily have non-singular OPEs with everything due to the fact that we are free to collide in the $z$ plane at finite separation in $t$ (up to $Q_{HT}$-exact terms), the OPE must then be non-singular due to locality. From this perspective, we identify $\frac{1}{2}\epsilon^{\beta\alpha} : \theta_\alpha \theta_\beta :$ with the local operator $\Theta_A| = \bar{\lambda} V_1|$.

### 4.1.2 Deformation to the $B$-twist

In order to deform to the $B$ twist we first need to set $\nu \to 1$; this ensures the operator $\phi^2 V_{-1}$ has spin 1 and has the effect of making all of the above generators spin 1. The $B$-twist is then realized by introducing the superpotential $W = -\phi^2 V_{-1}$. Note that this superpotential explicitly breaks the topological flavor symmetry.

The effects induced by deforming by a superpotential are described in Section 5 of [32], and we propose a mild generalization holds here. Note that this superpotential and its first derivatives vanish on the boundary due to the Dirichlet boundary conditions $\phi| = 0$ on the chiral multiplet. It follows, *cf.* Eq. (5.28) of *loc. cit.*, that this superpotential introduces an OPE of $\theta_+$ with itself of the form

$$\bar{\lambda}(z)\bar{\lambda}(w) \sim \frac{\partial_\phi^2 W(w)}{z - w} \quad \rightsquigarrow \quad \theta_-(z)\theta_-(w) \sim \frac{-2f(w)}{z - w}. \tag{49}$$

The remaining OPEs uniquely determined by associativity, *i.e.* they are uniquely determined by imposing that the commutators of the modes of these generators satisfy the Jacobi identity. For example, consider the OPE of $\theta_+$ and $\theta_-$ or, equivalently, the anti-commutator $\{\theta_{+,n}, \theta_{-,m}\}$. Using the relation $\theta_{+,n} = -[e_{n-m}, \theta_{-,m}]$ and $[\theta_{-,m}, \theta_{-,n}] = -2f_{n+m}$, it follows that

$$\begin{aligned}
\{\theta_{+,n}, \theta_{-,m}\} &= -\{[e_{n-m}, \theta_{-,m}], \theta_{-,m}\} \\
&= [\{\theta_{-,m}, \theta_{-,m}\}, e_{n-m}] - \{[\theta_{-,m}, e_{n-m}], \theta_{-,m}\} \\
&= 2[e_{n-m}, f_{2m}] - \{\theta_{+,n}, \theta_{-,m}\},
\end{aligned} \tag{50}$$

from which we conclude this anti-commutator is given by

$$\{\theta_{+,n}, \theta_{-,m}\} = [e_{n-m}, f_{2m}] = h_{n+m} + 2n\delta_{n+m,0}, \tag{51}$$

corresponding to the OPE

$$\theta_+(z)\theta_-(w) \sim \frac{2}{(z-w)^2} + \frac{h(w)}{z-w}. \tag{52}$$

The OPE of $\theta_+$ with itself can be computed similarly, leading to

$$\theta_+(z)\theta_+(w) \sim \frac{2e(w)}{z-w}. \tag{53}$$

We immediately see that the algebra of boundary local operators can be identified with an affine $\mathfrak{osp}(1|2)$ current algebra at level 1. More precisely, we find that the algebra of boundary local operators is identified with the simple quotient of the universal affine current algebra:

$$\mathcal{V} = L_1(\mathfrak{osp}(1|2)). \tag{54}$$

This vertex algebra is sometimes denoted $B_{0|1}(5,1)$, cf. [59,60]. As the generating currents all have spin 1, we identify the boundary stress tensor with the Sugawara stress tensor for $\mathfrak{osp}(1|2)$

$$T(z) = \frac{1}{5}\Big(\frac{1}{2}:h^2:+:ef:+:fe:+\frac{1}{2}\epsilon^{\beta\alpha}:\theta_\alpha\theta_\beta:\Big), \tag{55}$$

which has central charge $c = \frac{2}{5}$. It is straightforward to verify the above half-index $\mathbb{II}_B$ exactly reproduces the vacuum character of $L_1(\mathfrak{osp}(1|2))$ (up to a factor of the modular anomaly $q^{-\frac{c}{24}} = q^{-\frac{1}{60}}$):

$$
\begin{aligned}
\mathbb{II}_B(q;y) &= (q;q)_\infty^{-1} \sum_{\mathfrak{m}} q^{\mathfrak{m}^2} y^{2\mathfrak{m}} (y^{-1}q^{1-\mathfrak{m}};q)_\infty \\
&= (q;q)_\infty^{-1} \sum_{\mathfrak{m}} \sum_{\ell \geq 0} (-1)^\ell q^{\mathfrak{m}(\mathfrak{m}-\ell)+\frac{1}{2}\ell(\ell+1)} (q;q)_\ell^{-1} y^{2\mathfrak{m}-\ell} \\
&= \Big(\sum_{\mathfrak{m}} \frac{q^{\mathfrak{m}^2} y^{2\mathfrak{m}}}{(q;q)_\infty}\Big)\Big(\sum_{n\geq 0} \frac{q^{n(n+1)}}{(q;q)_{2n}}\Big) - \Big(\sum_{\mathfrak{m}} \frac{q^{\mathfrak{m}^2+\mathfrak{m}} y^{2\mathfrak{m}+1}}{(q;q)_\infty}\Big)\Big(\sum_{n\geq 0} \frac{q^{(n+1)^2}}{(q;q)_{2n+1}}\Big) \\
&= q^{\frac{1}{60}} \chi[L_1(\mathfrak{osp}(1|2))].
\end{aligned}
\tag{56}
$$

In the first line we used the $q$-binomial theorem; in the second line we split the sum over $\ell$ into even and odd parts and shifted $\mathfrak{m} \to \mathfrak{m} + \lfloor \frac{\ell}{2} \rfloor$.

## 4.2 Simple modules and fusion rules

Although $\mathcal{V}$ is not quite a Virasoro minimal model, it is quite close to both $M(3,5)$ and $M(2,5)$, albeit in quite different ways. As shown in [61] (see also [62,63]), the coset of the above $\mathfrak{osp}(1|2)$ current algebra by the subalgebra of $\mathfrak{sl}(2)$ currents is actually a minimal model:

$$M(3,5) \simeq \frac{L_1(\mathfrak{osp}(1|2))}{L_1(\mathfrak{sl}(2))}. \tag{57}$$

This Virasoro subalgebra is precisely generated by the bilinear $\frac{1}{2}\epsilon^{\beta\alpha}:\theta_\alpha\theta_\beta:=\Theta_A|$. As a module for $L_1(\mathfrak{sl}(2)) \otimes M(3,5)$, we have the following branching rules, cf. Eq. (14) of [63]:

$$L_1(\mathfrak{osp}(1|2)) \simeq \mathcal{L}_{1,0} \otimes V_{1,1}^{(3,5)} \oplus \Pi(\mathcal{L}_{2,0} \otimes V_{1,4}^{(3,5)}). \tag{58}$$

where $\Pi$ denotes a shift in fermionic parity, $\mathcal{L}_{i,0}$ ($i = 1,2$) are the simple modules for $L_1(\mathfrak{sl}(2))$, and $V_{r,s}^{(3,5)}$ ($r = 1,2$, $s = 1,...,4$ with $V_{r,s}^{(3,5)} \simeq V_{3-r,5-s}^{(3,5)}$ – we will often take $r = 1$) are the simple modules for $M(3,5)$. More precisely, $L_1(\mathfrak{osp}(1|2))$ is a $\mathbb{Z}_2$ simple current extension of $L_1(\mathfrak{sl}(2)) \otimes M(3,5)$, where the $\mathbb{Z}_2$ simple current we extend by is precisely $\Pi(\mathcal{L}_{2,0} \otimes V_{1,4}^{(3,5)})$ and is identified with the module generated by the action of $L_1(\mathfrak{sl}(2)) \otimes M(3,5)$ on the fermions $\theta_\pm$.

The realization of $L_1(\mathfrak{osp}(1|2))$ as an extension of the affine algebra $L_1(\mathfrak{sl}(2))$ and the minimal model $M(3,5)$ implies that its category of modules can be determined from those of these more familiar pieces [64]; this perspective on the category of $L_1(\mathfrak{osp}(1|2))$ modules was taken in [60] and we review it below, see e.g. [59] for a complementary perspective.

For ease of notation, denote $\mathcal{U} = L_1(\mathfrak{sl}(2)) \otimes M(3,5)$. Because $L_1(\mathfrak{sl}(2))$ and $M(3,5)$ are rational, any $\mathcal{U}$ module can be realized as a direct sum of simple objects of the form $\mathcal{L}_{i,0} \otimes V_{r,s}^{(3,5)}$. The fermionic module $\mathcal{X} = \Pi(\mathcal{L}_{2,0} \otimes V_{1,4}^{(3,5)})$ is the $\mathbb{Z}_2$ simple current we use to extend $\mathcal{U}$ to $\mathcal{V}$. For any module $M$ of $\mathcal{U}$, we can induce a not-necessarily-local module for $\mathcal{V}$ via the functor $\mathcal{F}(M) = \mathcal{V} \times M = M \oplus \mathcal{X} \times M$. Not all $M$ will induce a genuine, i.e. local or untwisted, module for $\mathcal{V}$: only those modules $M$ that have trivial monodromy with $\mathcal{X}$ will induce modules for $\mathcal{V}$. The modules $M$ and $\mathcal{X} \times M$ will result in the same module under $\mathcal{F}$, so we must

identify modules of $\mathcal{U}$ that differ by fusion with $\mathcal{X}$. It is worth emphasizing that simple current extensions are a boundary VOA manifestation of gauging a 1-form global symmetry in the bulk 3d QFT. Namely, gauging such a 1-form symmetry removes gauge non-invariant line operators (those having nontrivial monodromy with $\mathcal{X}$) and identifies line operators that differ by the action of the symmetry (those that differ by fusion with $\mathcal{X}$). We write this as follows:

$$L_1(\mathfrak{osp}(1|2)) \simeq \big(L_1(\mathfrak{sl}(2)) \otimes M(3,5)\big)\big/ \mathbb{Z}_2^{(1)}. \tag{59}$$

The desired monodromies are determined by those of $L_1(\mathfrak{sl}(2))$ and $M(3,5)$:

$$\mathcal{M}(\mathcal{L}_{2,0}, \mathcal{L}_{i,0}) = (-1)^{i+1}\mathrm{Id}_{\mathcal{L}_{2,0}\times\mathcal{L}_{i,0}}, \qquad \mathcal{M}(V_{1,4}^{(3,5)}, V_{1,s}^{(3,5)}) = (-1)^{s+1}\mathrm{Id}_{V_{1,4}^{(3,5)}\times V_{1,s}^{(3,5)}}. \tag{60}$$

We see that $\mathcal{L}_{i,0} \otimes V_{1,s}^{(3,5)}$ induces a local module if $i+s$ is even, leaving us with the following $\mathcal{V}$ modules:

$$\begin{aligned}
\mathcal{F}(\mathcal{L}_{1,0} \otimes V_{1,1}^{(3,5)}) &= \mathcal{L}_{1,0} \otimes V_{1,1}^{(3,5)} \oplus \Pi(\mathcal{L}_{2,0} \otimes V_{1,4}^{(3,5)}) =: \mathbf{1}, \\
\mathcal{F}(\mathcal{L}_{2,0} \otimes V_{1,2}^{(3,5)}) &= \mathcal{L}_{2,0} \otimes V_{1,2}^{(3,5)} \oplus \Pi(\mathcal{L}_{1,0} \otimes V_{1,3}^{(3,5)}) =: \Pi\mathbf{M}, \\
\mathcal{F}(\mathcal{L}_{1,0} \otimes V_{1,3}^{(3,5)}) &= \mathcal{L}_{1,0} \otimes V_{1,3}^{(3,5)} \oplus \Pi(\mathcal{L}_{2,0} \otimes V_{1,2}^{(3,5)}) =: \mathbf{M}, \\
\mathcal{F}(\mathcal{L}_{2,0} \otimes V_{1,4}^{(3,5)}) &= \mathcal{L}_{2,0} \otimes V_{1,4}^{(3,5)} \oplus \Pi(\mathcal{L}_{1,0} \otimes V_{1,1}^{(3,5)}) =: \Pi\mathbf{1}.
\end{aligned} \tag{61}$$

The remaining $\mathcal{L}_{i,0} \otimes V_{1,s}^{(3,5)}$ induce $\mathbb{Z}_2$-twisted modules where $\theta_\pm$ are half-integer moded. A priori, because our simple current extension is by $\mathcal{X} = \Pi(\mathcal{L}_{2,0}\otimes V_{1,4}^{(3,5)})$, we should also consider modules induced from the parity reversed modules $\Pi(\mathcal{L}_{i,0} \otimes V_{1,s}^{(3,5)})$, but those are equivalent to the above by fusion with $\mathcal{X}$. We note that the (super)character of $\mathbf{M}$ precisely matches the half-index in the presence of a Wilson line of charge $-1$ (up to a factor of the modular anomaly $q^{\frac{1}{5}-\frac{c}{24}} = q^{\frac{11}{60}}$):

$$\begin{aligned}
\mathbb{I}_B[\mathcal{W}_{-1}] &= -(q;q)_\infty^{-1} \sum_{\mathfrak{m}\in\mathbb{Z}} q^{\mathfrak{m}^2-\mathfrak{m}} y^{2\mathfrak{m}-1}(y^{-1}q^{1-\mathfrak{m}};q)_\infty \\
&= \bigg(\sum_{\mathfrak{m}} \frac{q^{\mathfrak{m}^2}y^{2\mathfrak{m}}}{(q;q)_\infty}\bigg)\bigg(\sum_{n\geq 0} \frac{q^{n^2+n}}{(q;q)_{2n+1}}\bigg) - \bigg(\sum_{\mathfrak{m}} \frac{q^{\mathfrak{m}^2+\mathfrak{m}}y^{2\mathfrak{m}+1}}{(q;q)_\infty}\bigg)\bigg(\sum_{n\geq 0} \frac{q^{n^2}}{(q;q)_{2n}}\bigg) \\
&= q^{-\frac{11}{60}}\chi[\mathbf{M}].
\end{aligned} \tag{62}$$

An important structural property of the induction functor $\mathcal{F}$ is that fusion of $\mathcal{V}$ modules is induced from fusion of the underlying $\mathcal{U}$ modules:

$$M_i \times M_j = \bigoplus N_{ij}^k M_k \qquad \Rightarrow \qquad \mathcal{F}(M_i) \times \mathcal{F}(M_j) = \bigoplus N_{ij}^k \mathcal{F}(M_k). \tag{63}$$

With this formula, we find that the fusion of $\mathbf{M}$ and itself is given by

$$\mathbf{M} \times \mathbf{M} = \mathcal{F}(\mathcal{L}_{1,0} \otimes V_{1,3}^{(3,5)} \times \mathcal{L}_{1,0} \otimes V_{1,3}^{(3,5)}) = \mathcal{F}(\mathcal{L}_{1,0} \otimes V_{1,1}^{(3,5)}) \oplus \mathcal{F}(\mathcal{L}_{1,0} \otimes V_{1,3}^{(3,5)}) = \mathbf{1} \oplus \mathbf{M}. \tag{64}$$

The remaining fusion rules follow from the interplay between fusion and parity shift $\Pi$:

$$(\Pi M_1) \times M_2 = \Pi(M_1 \times M_2) = M_1 \times (\Pi M_2). \tag{65}$$

For example, we also have

$$\mathbf{M} \times \Pi\mathbf{M} = \Pi\mathbf{1} \oplus \Pi\mathbf{M} = \Pi\mathbf{M} \times \mathbf{M}, \qquad \Pi\mathbf{M} \times \Pi\mathbf{M} = \mathbf{1} \oplus \mathbf{M}. \tag{66}$$

so that the fusion ring generated by $\Pi\mathbf{M}$ contains all of the simple modules $\mathbf{1}$, $\Pi\mathbf{1}$, $\mathbf{M}$, and $\Pi\mathbf{M}$. We note that the fusion ring generated by $\mathbf{M}$ precisely matches that of the Lee-Yang minimal model $M(2,5)$.

### 4.3 Modular transformations of supercharacters

In this subsection we turn to the modular transformation properties of the modules found above; as with fusion, these can be derived from those of the $\mathcal{U} = L_1(\mathfrak{sl}(2)) \otimes M(3,5)$ subalgebra, *cf.* [64]. Note that the supercharacters of a module and its parity shift agree with one another up to a factor of $-1$, $\chi[M] = -\chi[\Pi M]$. Correspondingly, we will restrict our attention to the modules $\mathbf{1}$ and $\mathbf{M}$.[4]

The fact that the vertex algebras $\mathcal{V}$ and $\mathcal{U}$ have integral conformal dimensions implies that the action of $T$ just measures the difference between the lowest conformal dimension and $\frac{c}{24} = \frac{1}{60}$, mod $\mathbb{Z}$; the lowest conformal dimensions are $0$ and $\frac{1}{5}$ for $\mathbf{1}$ and $\mathbf{M}$, respectively, whence the $T$-matrix is given by

$$T = \begin{pmatrix} e^{2\pi i \frac{-1}{60}} & 0 \\ 0 & e^{2\pi i \frac{11}{60}} \end{pmatrix}. \tag{67}$$

We can determine the modular $S$-matrix by using the $S$-matrices of $L_1(\mathfrak{sl}(2))$ and $M(3,5)$ together with the decompositions of $\mathbf{1}$ and $\mathbf{M}$ into $\mathcal{U}$ modules presented above. For example, consider the vacuum module $\mathbf{1} = \mathcal{L}_{1,0} \otimes V_{1,1}^{(3,5)} \oplus \Pi(\mathcal{L}_{2,0} \otimes V_{1,4}^{(3,5)})$; applying the $S$ transformations for $L_1(\mathfrak{sl}(2))$ and $M(3,5)$ we find

$$\chi[\mathbf{1}] = \chi[\mathcal{L}_{1,0}]\chi[V_{1,1}^{(3,5)}] - \chi[\mathcal{L}_{2,0}]\chi[V_{1,4}^{(3,5)}]$$
$$\to \frac{1}{2\sqrt{2}}\left(\chi[\mathcal{L}_{1,0}] + \chi[\mathcal{L}_{2,0}]\right)\left(\sqrt{1+\frac{1}{\sqrt{5}}}(\chi[V_{1,1}^{(3,5)}] - \chi[V_{1,4}^{(3,5)}]) + \sqrt{1-\frac{1}{\sqrt{5}}}(\chi[V_{1,2}^{(3,5)}] - \chi[V_{1,3}^{(3,5)}])\right)$$
$$+ \frac{1}{2\sqrt{2}}\left(\chi[\mathcal{L}_{1,0}] - \chi[\mathcal{L}_{2,0}]\right)\left(\sqrt{1+\frac{1}{\sqrt{5}}}(\chi[V_{1,1}^{(3,5)}] + \chi[V_{1,4}^{(3,5)}]) - \sqrt{1-\frac{1}{\sqrt{5}}}(\chi[V_{1,2}^{(3,5)}] + \chi[V_{1,3}^{(3,5)}])\right)$$
$$= \sqrt{\tfrac{1}{2}(1+\tfrac{1}{\sqrt{5}})}\chi[\mathbf{1}] - \sqrt{\tfrac{1}{2}(1-\tfrac{1}{\sqrt{5}})}\chi[\mathbf{M}]. \tag{68}$$

Similarly, we find $\chi[\mathbf{M}] \to -\sqrt{\tfrac{1}{2}(1-\tfrac{1}{\sqrt{5}})}\chi[\mathbf{1}] - \sqrt{\tfrac{1}{2}(1+\tfrac{1}{\sqrt{5}})}\chi[\mathbf{M}]$, leading to the following $S$-matrix:

$$S = \begin{pmatrix} \sqrt{\tfrac{1}{2}(1+\tfrac{1}{\sqrt{5}})} & -\sqrt{\tfrac{1}{2}(1-\tfrac{1}{\sqrt{5}})} \\ -\sqrt{\tfrac{1}{2}(1-\tfrac{1}{\sqrt{5}})} & -\sqrt{\tfrac{1}{2}(1+\tfrac{1}{\sqrt{5}})} \end{pmatrix}. \tag{69}$$

It is straightforward to verify these $S$- and $T$-matrices satisfy the defining relations $S^2 = (ST)^3 = 1$ of $PSL(2,\mathbb{Z})$.

It is possible to reproduce these modular data from the partition functions of $\mathcal{T}_{\min}$ calculated on Seifert manifolds. If $M_{g,p}$ is a degree-$p$ circle bundle over a closed Riemann surface of genus $g$, the partition function can be written as

$$Z_{M_{g,p}} = \sum_{P(u_*)=1} \mathcal{H}^{g-1}(u_*)\mathcal{F}^p(u_*), \tag{70}$$

where $\mathcal{H}(u_*)$ and $\mathcal{F}(u_*)$ are certain functions that can be computed from the twisted effective superpotential $\mathcal{W}(u)$ of the UV gauge theory, evaluated at the solutions to the Bethe-equations,

$$P(u) = \exp\left[2\pi i \frac{\partial \mathcal{W}(u)}{\partial u}\right] = 1. \tag{71}$$

We find that, in the B-twist limit, $\mathcal{H}(u_*)$ and $\mathcal{F}(u_*)$ reproduce the data $\{S_{0\alpha}^{-2}\}$ and $\{T_{\alpha\alpha}\}$ of Eqs. (69) and (67) respectively, up to an overall phase factor.

---

[4]We could equivalently consider $\mathbf{1}$ and $\Pi\mathbf{M}$ by conjugating the following $S$- and $T$-matrices by the diagonal matrix diag$(1,-1)$, corresponding to the change of basis from $\{\chi[\mathbf{1}], \chi[\mathbf{M}]\}$ to $\{\chi[\mathbf{1}], \chi[\Pi\mathbf{M}]\}$.

We note that the above $S$- and $T$-matrices are quite close to those of the minimal model $M(2,5)$. Namely, they differ from those of $M(2,5)$ only by conjugation:

$$T = \begin{pmatrix} 0 & -1 \\ 1 & 0 \end{pmatrix} T_{M(2,5)} \begin{pmatrix} 0 & 1 \\ -1 & 0 \end{pmatrix}, \qquad S = \begin{pmatrix} 0 & -1 \\ 1 & 0 \end{pmatrix} S_{M(2,5)} \begin{pmatrix} 0 & 1 \\ -1 & 0 \end{pmatrix}. \tag{72}$$

This conjugation is consistent with the fact that the $y \to 1$ limit of the supercharacters of $\mathbf{1}$ and $\Pi \mathbf{M}$ precisely agree with those of the $M(2,5)$ modules $V_{1,2}^{(2,5)}$ and $V_{1,1}^{(2,5)}$, respectively.[5]

## 4.4 A boundary condition for $M(2,5)$

We note that the above analysis has been concerned with *right* boundary conditions, *i.e.* considering bulk theories living on $\mathbb{C} \times \mathbb{R}_{\leq 0}$, *cf.* Section 4.4 of [51]. There are equally good half-BPS $(0,2)$ boundary conditions when we consider theories on $\mathbb{C} \times \mathbb{R}_{\geq 0}$. We can compute half-indices counting local operators on left boundary conditions exactly as before. In fact, we find that there is a left boundary condition whose half-index precisely reproduces the vacuum character of $M(2,5)$.

We start by considering a Neumann boundary condition $(\mathcal{N}, N)$. This does not give a well-defined boundary condition due to a gauge anomaly: the bulk contribution to the anomaly polynomial is the negative of the one appearing in Eq. (32). We can cancel all gauge anomalies by introducing two boundary fermions; one of gauge charge 1 and the other of gauge charge $-1$ and $U(1)_H$ charge 1.[6] We shall denote this dressed Neumann boundary condition Neu. A particularly important point is that the $U(1)_H$ symmetry is anomalous on the boundary; as this symmetry is necessary for defining the twisting homomorphism the $A$-twist, we see that this boundary condition is only compatible with the $B$-twist.

The $B$-twisted half-index counting local operators on Neu realizes the vacuum character of $M(2,5)$ (up to an overall factor of the modular anomaly):

$$\begin{aligned}
I\!I_{\text{Neu}}(q) &= (q;q)_\infty \oint \frac{dy}{2\pi i y} \frac{(qy;q)_\infty (y^{-1};q)_\infty (qy^{-1};q)_\infty (y;q)_\infty}{(y;q)_\infty} \\
&= (q;q)_\infty \sum_{m,n \geq 0} \frac{q^{m+m^2+mn+n^2}}{(q;q)_{m+n}(q;q)_m(q;q)_n} \\
&= \sum_{m \geq 0} \frac{q^{m+m^2}}{(q;q)_m} = q^{\frac{11}{60}} \chi_{\alpha=0}^{M(2,5)}(q).
\end{aligned} \tag{73}$$

In going from the first line to the second, we use the $q$-binomial theorem and extracted the terms proportional to $y^0$; in going from the second to the third we used the identity[7]

$$(q;q)_\infty^{-1} = \sum_{n \geq 0} \frac{q^{mn+n^2}}{(q;q)_{m+n}(q;q)_n}. \tag{74}$$

---

[5]Our $S$-matrix agrees with the one appearing in Eq. (3.11) of [5], but the $T$-matrix is slightly different:

$$T_{GKLSY} = e^{-2\pi i/60} T^{-1}, \qquad S_{GKLSY} = S.$$

This difference in the $T$-matrix is arises because the $T$-matrices presented in [5] are only determined up to an overall phase factor; indeed, the matrices $T_{GKLSY}$ and $S_{GKLSY}$ do not satisfy $(S_{GKLSY} T_{GKLSY})^3 = 1$.

[6]The field strength $\mathbf{r} - \mathbf{f}_{\text{top}}$ couples to $U(1)_C$ whereas $\mathbf{r} + \mathbf{f}_{\text{top}}$ couples to $U(1)_H$:

$$\mathbf{f}_{\text{top}} A + \mathbf{r} R = (\mathbf{r} + \mathbf{f}_{\text{top}}) R_C + (\mathbf{r} - \mathbf{f}_{\text{top}}) R_H,$$

where we used $A = -S = R_C - R_H$ and $R = R_C + R_H$.

[7]This identity can be seen as a consequence of the Jacobi triple product/bosonization identity

$$(-yq;q)_\infty (-y^{-1};q) = \sum_{m \in \mathbb{Z}} \frac{q^{\frac{1}{2}m(m+1)} y^m}{(q;q)_\infty},$$

In the same fashion, we can compute the half-index for the Wilson line $\mathcal{W}_{-1}$ (together with the background $R$-symmetry Wilson line) to find the character of the other minimal model

$$
\begin{aligned}
I\!I_{\text{Neu}}(q)[\mathcal{W}_{-1}] &= (q;q)_\infty \oint \frac{\mathrm{d}y}{2\pi i y} \frac{(qy;q)_\infty (y^{-1};q)_\infty (qy^{-1};q)_\infty (y;q)_\infty}{(y;q)_\infty}(-y) \\
&= \sum_{m\geq 0} \frac{q^{m^2}}{(q;q)_m} = q^{-\frac{1}{60}} \chi^{M(2,5)}_{\alpha=1}(q).
\end{aligned}
\tag{75}
$$

Although we do not know the precise form of how to realize the $B$-twist deformation of Neu, the above evidence suggests that the algebra of local operators on Neu surviving the $B$-twist can be identified with the $M(2,5)$ minimal model. We now explain how $M(2,5)$ is related to our choice of boundary fermions. First note that the VOA of boundary fermions, identified with two copies of the $bc$ ghost system, has an $L_1(\mathfrak{sl}(2))$ symmetry rotating the two pairs of fermions into one another and whose coset inside the free-fermion algebra is another copy of $L_1(\mathfrak{sl}(2))$. This is an avatar of the exceptional isomorphism $\mathfrak{so}(4) \sim \mathfrak{sl}(2) \oplus \mathfrak{sl}(2)$, where the $\mathfrak{so}(4)$ currents are realized by the bilinears of the fermions. Importantly, the Urod theorem of [65] says that this latter $L_1(\mathfrak{sl}(2))$ contains a copy of $M(2,5)$ and hence we have an embedding of $M(2,5)$ into the VOA of our boundary free fermions. The remarkable fact is that the commutant of $M(2,5)$ inside of our boundary fermions is precisely $L_1(\mathfrak{osp}(1|2))$ and, moreover, they are mutual commutants of one another:

$$
\frac{\text{bc}^{\otimes 2}}{L_1(\mathfrak{osp}(1|2))} \simeq M(2,5) \hookrightarrow \text{bc}^{\otimes 2} \hookleftarrow L_1(\mathfrak{osp}(1|2)) \simeq \frac{\text{bc}^{\otimes 2}}{M(2,5)}.
\tag{76}
$$

Indeed, $M(2,5) \otimes L_1(\mathfrak{osp}(1|2)) \hookrightarrow \text{bc}^{\otimes 2}$ is a conformal embedding:

$$
c_{M(2,5)} + c_{L_1(\mathfrak{osp}(1|2))} = -\tfrac{22}{5} + \tfrac{2}{5} = -4 = 2c_{\text{bc}}.
\tag{77}
$$

As a module for $M(2,5) \otimes L_1(\mathfrak{osp}(1|2))$, the two bc ghost systems decompose as

$$
\text{bc}^{\otimes 2} = V^{(2,5)}_{1,1} \otimes \mathbf{1} \oplus V^{(2,5)}_{1,2} \otimes \mathbf{M}.
\tag{78}
$$

We propose that the $B$ twist deforms the algebra of local operators on Neu to the coset of the boundary fermions by their $L_1(\mathfrak{osp}(1|2))$ subalgebra.

We can view the appearance of $M(2,5)$ on a left boundary condition and $L_1(\mathfrak{osp}(1|2))$ on a right boundary condition as a version of level-rank duality. In the context of, *e.g.*, $SU(n)_k$ Chern-Simons theory, a holomorphic Dirichlet right boundary condition leads to the usual affine current algebra $L_k(\mathfrak{sl}(n))$. There is a left holomorphic boundary condition realized by dressing a Neumann boundary condition by $k$ fundamental chiral fermions, leading to the commutant of the free fermion VOA by their $L_k(\mathfrak{sl}(n))$ subalgebra, which is itself isomorphic to the affine current algebra $L_n(\mathfrak{gl}(k))$. These algebras were shown to have equivalent categories of modules, up to a reversal of the braiding [66]; see also the earlier physical work [67] and the more recent [68].

Physically, modules for either VOA can be used to model line operators in the bulk and hence their categories of modules must be equivalent as abelian categories; as one VOA is on

---

by applying the $q$-binomial theorem to the left-hand side. Explicitly, the $q$-binomial theorem gives us

$$
\begin{aligned}
(-yq;q)_\infty (-y^{-1};q) &= \sum_{k,l\geq 0} \frac{q^{\frac{1}{2}k(k+1)+\frac{1}{2}l(l-1)} y^{k-l}}{(q;q)_k (q;q)_l} \\
&= \sum_{\text{m}\geq 0} q^{\frac{1}{2}\text{m}(\text{m}+1)} y^{\text{m}} \sum_{l\geq 0} \frac{q^{l^2+l\text{m}}}{(q;q)_{l+\text{m}}(q;q)_l} + \sum_{\text{m}<0} q^{\frac{1}{2}\text{m}(\text{m}+1)} y^{\text{m}} \sum_{k\geq 0} \frac{q^{k^2-k\text{m}}}{(q;q)_k(q;q)_{k-\text{m}}},
\end{aligned}
$$

from which the claimed identity follows by equating the coefficients with non-negative powers of $y$.

the left and the other on the right, the natural braiding of modules of the VOA is reversed from one another, leading to a braid-reversed equivalence of categories. In the context of $SU(n)_k$ Chern-Simons theory, this relation between VOAs is often rephrased in terms of a relation between Chern-Simons theories, where the reversal of the braiding is realized by negating the level as

$$SU(n)_k \simeq U(k)_{-n,-n}. \tag{79}$$

More generally, whenever one has a commuting pair inside a VOA with trivial category of modules, such as $M(2,5) \otimes L_1(\mathfrak{osp}(1|2))$ inside $\mathrm{bc}^{\otimes 2}$, there is necessarily a braid-reversed equivalence between the two VOAs (under to some technical assumptions about the relevant vertex tensor categories which hold in this setting) [69]. The decomposition in Eq. (78) dictates how these modules are matched under this equivalence.

# 5 Higher level VOAs

A first-principles derivation of a boundary VOA $\mathcal{V}_r$ for the higher-level theory $\mathcal{T}_r$ is much harder to implement than the level 1 case. In brief, this is due to the form of the deformations proposed by [27]. Instead, we present indirect evidence for the following proposal: the $B$ twist of $\mathcal{T}_r$ admits

$$\mathcal{V}_r = L_r(\mathfrak{osp}(1|2)) \tag{80}$$

as a boundary VOA. We compare the fusion rules of $L_r(\mathfrak{osp}(1|2))$ modules and the modular $S$- and $T$-transformations of their characters to the analysis of [27], finding compatible results. One interesting consequence of our proposal is a fermionic sum representation of $L_r(\mathfrak{osp}(1|2))$ characters, see Eqs. (82) and (95).

## 5.1 A deformable boundary condition for $\mathcal{T}_r$

We start by identifying a boundary condition for $\mathcal{T}_r$ that generalizes the Dirichlet boundary condition Dir studied in Section 4. In particular, we will focus our attention on (right) $\mathcal{N} = (0,2)$ boundary conditions that impose Dirichlet boundary conditions on the vector multiplets and the chiral multiplets. Such a boundary condition is specified by the values $\phi_i|$ of the chiral multiplet scalars on the boundary. The half-index counting local operators on such a boundary condition (with the spins relevant for the $B$ twist) can be obtained by specializing the half-index counting local operators on the Dirichlet boundary condition where all of the scalars vanish on the boundary:

$$\mathbb{I}(y_i; q) = (q;q)_\infty^{-r} \sum_{\vec{\mathfrak{m}} \in \mathbb{Z}^r} q^{\frac{1}{2} \vec{\mathfrak{m}}^T K_r \vec{\mathfrak{m}}} \vec{y}^{K_r \vec{\mathfrak{m}}} (y_1^{-1} q^{1-\mathfrak{m}_1}; q)_\infty \dots (y_r^{-1} q^{1-\mathfrak{m}_r}; q)_\infty, \tag{81}$$

where $y_i$ are fugacities the $U(1)_{i,\partial}$. A nonzero value for $\phi_i|$ breaks the $U(1)_{i,\partial}$ symmetry, forcing us to specialize the above index as $y_i \to 1$, *cf.* Section 3.2.2 of [51]. We note that this expression only contains non-negative powers of $q$, and the coefficient of $q^0$ is 1.

### 5.1.1 Superpotential constraints

As described in Section 3, the allowed boundary conditions are constrained by the fact that they must be compatible with the bulk superpotential. The superpotential relevant for the $B$ twist contains two parts. The first part comes from the bare monopoles $V_{\mathfrak{m}_1}, \dots, V_{\mathfrak{m}_{r-1}}$ which are argued by [27] to be necessary for the IR theory to exhibit an enhancement to $\mathcal{N} = 4$ supersymmetry. The second part is given by the dressed monopole operator $\phi_1^2 \dots \phi_r^2 V_{(-1,0,\dots,0)}$ and is used to deform (the $HT$ twist of) this theory to the $B$ twist of the IR SCFT.

The first constraints on the boundary condition we consider are those coming from the operator $\phi_1^2 \ldots \phi_r^2 V_{(-1,0,\ldots,0)}$. One constraint says that this operator must vanish when brought to the boundary: Dirichlet boundary conditions will break SUSY unless the superpotential vanishes at the boundary, *cf.* Section 2.3 of [51], and this breaking is related to the boundary condition being deformable [2,34]. This term in the superpotential suggests we should require that at least one of the scalars vanishes on the boundary. On the other hand, the superpotential should not vanish to a very high order. As described in [32], a non-vanishing $\partial_\phi W|$ leads to a differential on the fermions, a non-vanishing $\partial_\phi^2 W|$ leads to a singular OPE, and more generally non-vanishing components of the tensor of $n$th-order derivatives $\partial_\phi^n W|$ may lead to higher $L_\infty$ operations on the boundary vertex algebra. To ensure that the superpotential $\phi_1^2 \ldots \phi_r^2 V_{(-1,0,\ldots,0)}$ induces a suitable singular OPE and no higher operations,[8] we set $r-1$ of the scalars to non-zero values, with the remaining scalar forced to vanish on the boundary.[9]

Determining which $r-1$ scalars to set to nonzero values is a bit more delicate, but this is ultimately determined by the requirement that the bulk theory must also be deformed by a bare monopole superpotential. In order for this Dirichlet boundary condition to be compatible with this superpotential, it too must vanish on the boundary. In the context of ordinary 3d $\mathcal{N}=4$ gauge theories, a bulk bare monopole $V_{\vec{\mathfrak{m}}}$ of magnetic charge $\vec{\mathfrak{m}}$ vanishes on the boundary if $\mathfrak{m}_i Q_n^i > 0$ for all $n$, where $Q_n^i$ is the charge of the $n$th scalar given generic Dirichlet boundary conditions, *cf.* Eq. (4.10) of [70]. We propose this happens here as well, although we leave a detailed analysis to future work. This massive cancellation can be witnessed in the half-index: a fermion paired with a scalar of charge $Q^i$ given generic Dirichlet boundary conditions contributes $(q^{1-\mathfrak{m}_i Q^i}; q)_\infty$, which vanishes for all $\mathfrak{m}_i Q^i > 0$. With this observation in hand, we see that the bare monopoles $V_{\mathfrak{m}_i}$ all vanish on the boundary if we give the first $r-1$ of the scalars generic Dirichlet boundary conditions, and require the $r$th scalar vanish. It is worth noting that, with this criterion the fully generic Dirichlet boundary conditions studied by [27] are deformable to the $A$-twist. We think it would be quite interesting to understand more explicitly how the local operators on such a boundary condition realize the proposed minimal model $M(2, 2r+3)$.

In summary, we propose that the boundary condition $\mathrm{Dir}^{(r)}$ defined by

1) Dirichlet boundary conditions on the vector multiplets

2) generic/deformed Dirichlet boundary conditions on the first $r-1$ chiral multiplets $\phi_i| = b_i$, $i = 1, \ldots, r-1$, $b_i \in \mathbb{C}^\times$

3) Dirichlet boundary conditions for the $r$th chiral multiplets $\phi_r| = 0$

is deformable to the $B$ twist.

### 5.1.2 Evidence for $L_r(\mathfrak{osp}(1|2))$

With the boundary condition $\mathrm{Dir}^{(r)}$ in hand, we now turn to the algebra of boundary local operators. Unlike the level 1 case, we are unable to determine the algebra of local operators

---

[8]So long as holomorphic boundary condition is deformable, it should give rise to an equivalent (quasi-isomorphic) description of line operators in the bulk TQFT. The issue with higher operations is in the complexity required to extract the $E_2$ or braided-tensor structure of line operators from modules of the local operators; in a sense, we are simply seeking a strict model for the category of bulk line operators.

[9]A priori, the fact that the higher-order derivatives of $\phi_1^2 \ldots \phi_r^2 V_{(-1,0,\ldots,0)}$ do not vanish when we give $r-1$ of the scalars generic boundary values could lead to higher operations. Thankfully, this possibility is mitigated by the appearance of a differential that trivializes all but 1 boundary fermion. Schematically, the differential takes the form $QB = \mu| = \phi|\bar{\lambda}$ where $\mu$ is the current generating the $U(1)$ action on the chiral multiplet; this does not arise when $\phi| = 0$, but when $\phi| \neq 0$ it implements the breaking of the $U(1)_\partial$ symmetry as the generating current $B$ is no longer $Q$-closed.

directly. Instead, we present some indirect evidence supporting the proposal that this algebra of local operators realizes the simple affine VOA $L_r(\mathfrak{osp}(1|2))$.

Our main piece of evidence for this proposal comes from comparing half-indices and (super)characters of the simple modules of $L_r(\mathfrak{osp}(1|2))$, *cf.* [60]. The half-index counting local operators on $\text{Dir}^{(r)}$ is given by

$$\mathbb{I}^{(r)}(y;q) = \sum_{\mathfrak{m}\in\mathbb{Z}} \sum_{\vec{\mathfrak{n}}\in\mathbb{Z}^{r-1}_{\leq 0}} \frac{q^{\frac{1}{2}\vec{\mathfrak{n}}^T K_{r-1}\vec{\mathfrak{n}} + \mathfrak{m}\vec{k}_{r-1}\cdot\vec{\mathfrak{n}} + r\mathfrak{m}^2} y^{2\vec{k}_{r-1}\cdot\vec{\mathfrak{n}} + 2r\mathfrak{m}}(y^{-1}q^{1-\mathfrak{m}};q)_\infty}{(q;q)_\infty (q;q)_{\mathfrak{n}_1}\ldots(q;q)_{\mathfrak{n}_{r-1}}}\,, \tag{82}$$

where we write the matrix of Chern-Simons levels in block form as

$$K_r = \begin{pmatrix} K_{r-1} & 2\vec{k}_{r-1} \\ 2\vec{k}_{r-1}^T & 2r \end{pmatrix}\,. \tag{83}$$

Although we aren't able to show it analytically, we find that these are consistent with the vacuum character of $L_r(\mathfrak{osp}(1|2))$ by comparing $q$-expansions for low values of $r$. Showing such an equality would be quite interesting, and could be interpreted as a fermionic sum representation for this vacuum character.

We can use this half-index, together with the form of our superpotential deformations, to gain some insight into how various pieces of the $\mathfrak{osp}(1|2)$ current algebra fit together. First, the perturbative operator $B_r$ generating the (unbroken) boundary $U(1)_{r,\partial}$ symmetry has the necessary level to be the Cartan generator of an $\mathfrak{osp}(1|2)$ current algebra at level $r$:

$$B_r(z)B_r(w) \sim \frac{2r}{z-w}\,. \tag{84}$$

We note that the bare monopole $V_{(-1,0,\ldots,0)}|$ survives on $\text{Dir}^{(r)}$ due to its negative magnetic charge and has the necessary quantum numbers ($B_r$-charge $-2$ and spin 1) to be identified with the bosonic generator $f(z)$. Moreover, the term $\phi_1^2 \ldots \phi_r^2 V_{(-1,0,\ldots 0)}$ in the superpotential introduces a non-trivial OPE between the $r$th fermion $\overline{\lambda} = \overline{\lambda}_{r,-}|$ and itself:

$$\overline{\lambda}(z)\overline{\lambda}(w) \sim \frac{\phi_1^2|\ldots\phi_{r-1}^2|V_{(-1,0,\ldots,0)}|(w)}{z-w}\,, \tag{85}$$

which is precisely the form necessary for $\overline{\lambda}$ to be the fermionic generator $\theta_-$ of an $\mathfrak{osp}(1|2)$ current algebra. The bare monopole $V_{(0,\ldots,-1,1)}|$ also survives after deforming the Dirichlet boundary conditions and has the right quantum numbers to be identified with the other fermionic generator $\theta_+$. Finally, the half-index suggests the remaining bosonic generator $e(z)$ should be identified with a second boundary local operator of magnetic charge $(0,\ldots,-1,1)$.

Not only can we use the half-index to identify various local operators, if we assume it is indeed the vacuum character of a rational VOA then we can also use it to predict the central charge of that VOA using modular properties of the index. As noted in [27], the further specialization $y \to 1$ of this half-index realizes the character $\chi_{1,r+1}^{(2,2r+3)}$ of $M(2,2r+3)$

$$\chi_{1,r+1}^{(2,2r+3)}(q) = q^{h_{1,r+1}^{(2,2r+3)} - c^{(2,2r+3)}/24}\, \mathbb{I}^{(r)}(1;q)\,, \tag{86}$$

which follows from the fermionic sum representations due to [71–76]:

$$\chi_{1,n}^{(2,2r+3)}(q) = q^{h_{1,n}^{(2,2r+3)} - \frac{c^{(2,2r+3)}}{24}} \sum_{\vec{m}\in\mathbb{Z}^r_{\geq 0}} \frac{q^{\frac{1}{2}\vec{m}^T K_r \vec{m} + \vec{m}W_n^{(r)}}}{(q;q)_{m_1}\ldots(q;q)_{m_r}}\,, \tag{87}$$

where $c^{(2,2r+3)}$ is the central charge of $M(2,2r+3)$, $h_{1,n}^{(2,2r+3)}$ are the lowest scaling dimensions of its simple modules, and

$$W_n^{(r)} = (\overbrace{0,\ldots,0}^{n-1}, 1, 2, \ldots, r-n+1) \tag{88}$$

for $n = 1, \ldots, r+1$. If this is to be the vacuum character of a rational VOA, we can identify its central charge by matching modular anomalies:

$$-\frac{c_r}{24} = h_{1,r+1}^{(2,2r+3)} - \frac{c^{(2,2r+3)}}{24} \rightsquigarrow c_r = \frac{2r}{2r+3}. \tag{89}$$

Using that the (super)dimension and dual Coxeter number of $\mathfrak{osp}(1|2)$ are 1 and $\frac{3}{2}$, respectively, we see that this is precisely the Sugawara central charge of $L_r(\mathfrak{osp}(1|2))$.

## 5.2 Wilson lines and modules for $L_r(\mathfrak{osp}(1|2))$

Modules for $L_r(\mathfrak{osp}(1|2))$ can be described in the same way as for $r = 1$. We will again take the perspective of [60] and view it as an extension of $L_r(\mathfrak{sl}(2)) \otimes M(r+2, 2r+3)$. Equivalently, there is a coset realization of $M(r+2, 2r+3)$:

$$M(r+2, 2r+3) \simeq \frac{L_r(\mathfrak{osp}(1|2))}{L_r(\mathfrak{sl}(2))}. \tag{90}$$

The VOA $L_r(\mathfrak{osp}(1|2))$ decomposes as a module for the product $L_r(\mathfrak{sl}(2)) \otimes M(r+2, 2r+3)$ (taking fermionic parity into account) as

$$L_r(\mathfrak{osp}(1|2)) = \bigoplus_{n=1}^{r+1} \Pi^{n-1} \mathcal{L}_{n,0}^{(r)} \otimes V_{n,1}^{(r+2,2r+3)}. \tag{91}$$

As before, the decomposition into a sum of products of $L_r(\mathfrak{sl}(2))$ and $M(r+2, 2r+3)$ modules allows the representation theory of $L_r(\mathfrak{osp}(1|2))$ to be understood in terms of these pieces by way of induction [64].

The analysis of [27] predicted the number of simple objects and fusion rules of the category of line operators in $\mathcal{T}_r$, finding it should be the same as the minimal model $M(2, 2r+3)$. The VOA $L_r(\mathfrak{osp}(1|2))$ has $r+1$ simple modules (up shifts in parity), matching the number of simple objects of $M(2, 2r+3)$. Written in terms of $L_r(\mathfrak{sl}(2))$ and $M(r+2, 2r+3)$ modules, they take the following form:

$$\mathbf{M}_i = L_r(\mathfrak{osp}(1|2)) \times \left( \mathcal{L}_{1,0}^{(r)} \otimes V_{1,2i+1}^{(r+2,2r+3)} \right) = \bigoplus_{n=1}^{r+1} \Pi^{n-1} \mathcal{L}_{n,0}^{(r)} \otimes V_{n,2i+1}^{(r+2,2r+3)}, \tag{92}$$

$i = 0, \ldots r$; the vacuum module is identified with $\mathbf{M}_0$. In the notation of [60], these modules are denoted $\mathcal{A}_{2i+1,0}$. The fusion rules for the modules are induced from $L_r(\mathfrak{sl}(2))$ and $M(r+2, 2r+3)$. In particular, we find

$$\mathbf{M}_i \times \mathbf{M}_j = \bigoplus_{k=0}^{r} N_{2i+1,2j+1}^{(2r+3)2k+1} \mathbf{M}_k, \tag{93}$$

where the structure constants $N_{s,s'}^{(p)s''}$ are given by

$$N_{s,s'}^{(p)s''} = \begin{cases} 1 & \text{if } |s-s'|+1 \leq s'' \leq \min(s+s'-1, 2p-s-s'-1) \text{ and } s+s'+s'' \text{ is odd} \\ 0 & \text{else} \end{cases} \tag{94}$$

This precisely matches the fusion rules of $M(2, 2r+3)$ modules, where the identification of fusion rings is via $\mathbf{M}_i \leftrightarrow V_{1,2i+1}^{(2,2r+3)} \simeq V_{1,2(r+1-i)}^{(2,2r+3)}$.

We can again use the analysis of [27] to predict which bulk line operators are identified with which modules for $L_r(\mathfrak{osp}(1|2))$. Namely, we consider Wilson lines with charges $-W_n^r$ (dressed by a suitable $R$-symmetry Wilson line). The half-index counting local operators at the junction of such a Wilson line and $\mathrm{Dir}^{(r)}$ is given by

$$
\mathbb{I}_n^{(r)}(y;q) = \sum_{\mathfrak{m}\in\mathbb{Z}} \sum_{\vec{\mathfrak{n}}\in\mathbb{Z}_{\leq 0}^{r-1}} \frac{q^{\frac{1}{2}\vec{\mathfrak{n}}^T K_{r-1}\vec{\mathfrak{n}} + \mathfrak{m}\vec{k}_{r-1}\cdot\vec{\mathfrak{n}} + r\mathfrak{m}^2} y^{2\vec{k}_{r-1}\cdot\vec{\mathfrak{n}}+2r\mathfrak{m}}(y^{-1}q^{1-\mathfrak{m}};q)_\infty}{(q;q)_\infty(q;q)_{\mathfrak{n}_1}\ldots(q;q)_{\mathfrak{n}_{r-1}}}
$$
$$
\times \left((-1)^{r+1-n}(yq^{\mathfrak{m}})^{n-r-1}q^{-W_n^{(r-1)}\cdot\mathfrak{n}}\right), \tag{95}
$$

where $W_{r+1}^{(r-1)} = 0$. As observed by [27], the further specialization $y \to 1$ of these indices reproduces the remaining characters of the modules for $M(2, 2r+3)$ up to a sign:

$$
\chi_{1,n}^{(2,2r+3)}(q) = (-1)^{r+1-n}q^{h_{1,n}^{(2,2r+3)}-c^{(2,2r+3)}/24}\mathbb{I}_n^{(r)}(1;q). \tag{96}
$$

Using this observation, we can predict the minimal conformal weights appearing in the modules associated to these Wilson lines by comparing modular anomalies as we did for the central charge:

$$
h_n^{(r)} - \frac{c_r}{24} = h_{1,n}^{(2,2r+3)} - \frac{c^{(2,2r+3)}}{24} \rightsquigarrow h_n^{(r)} = \frac{(r+1-n)(r+2-n)}{2(2r+3)}. \tag{97}
$$

This precisely matches the minimal conformal weight appearing in the module $\mathbf{M}_{r+1-n}$. This leads us to the expectation that the half-index reproduces these (super)characters:

$$
\chi[\mathbf{M}_{r+1-n}](y;q) = q^{h_n^{(r)}-c_r/24}\mathbb{I}_n^{(r)}(y;q). \tag{98}
$$

We were unable to verify this identity analytically, but comparing the $q$-expansions of the unspecialized half-indices for low values of $r$ supports our expectation; we can again view these half-indices as fermionic sum representations for these characters.

The modular transformations of these characters can again be determined using the modular transformation properties of characters for $L_r(\mathfrak{sl}(2))$ and $M(r+2, 2r+3)$. We find that the resulting $S$- and $T$-matrices are simply obtained by conjugating those of $M(2, 2r+3)$ by a signed permutation:

$$
S = PS_{M(2,2r+3)}P^{-1}, \qquad T = PS_{M(2,2r+3)}P^{-1}, \qquad P = \begin{pmatrix} 0 & 0 & \ldots & (-1)^r \\ \vdots & \vdots & \ddots & \vdots \\ 0 & -1 & \ldots & 0 \\ 1 & 0 & \ldots & 0 \end{pmatrix}. \tag{99}
$$

The form of the signed permutation $P$ is compatible with the specialization $\chi[\mathbf{M}_{r+1-n}](1;q)$ being equal to $(-1)^{r+1-n}\chi_{1,n}^{(2,2r+3)}(q)$.

## Acknowledgments

We would like to thank Tudor Dimofte, Dongmin Gang, and especially Thomas Creutzig for useful discussions and comments.

**Funding information** The work of AEVF is supported by the EPSRC Grants EP/T004746/1 "Supersymmetric Gauge Theory and Enumerative Geometry" and EP/W020939/1 "3d N=4 TQFTs". NG is supported by funds from the Department of Physics and the College of Arts & Sciences at the University of Washington, Seattle. The work of HK is supported by the Ministry of Education of the Republic of Korea and the National Research Foundation of Korea grant NRF-2023R1A2C1004965. This research was supported in part by the International Centre for Theoretical Sciences (ICTS) for participating in the program - Vortex Moduli (code: ICTS/Vort2023/2).

# A  A QFT for Ising

In this appendix we propose an QFT and right boundary condition thereof that realizes the minimal model $M(3,4)$, *i.e.* (the symmetry algebra of) the critical 2d Ising model. The theory we consider is a 3d $\mathcal{N} = 2$ $U(1)^2$ gauge theory with UV Chern-Simons level

$$k_{UV} = \begin{pmatrix} -\frac{1}{2} & 2 \\ 2 & 0 \end{pmatrix}, \qquad (A.1)$$

coupled to a single chiral multiplet $\Phi$ of gauge charge $(1,0)$ and $R$-charge 1. Roughly speaking, the result of gauging with this $BF$-like Chern-Simons term is to couple to a $\mathbb{Z}_2$ gauge field acting on the chiral field as $\Phi \to -\Phi$, *cf.* [77,78]. We then deform this $\mathcal{N} = 2$ theory by a monopole superpotential of the form $\sim \Phi^2 V_{(0,-1)}$, where $V_{(m,n)}$ is the bare BPS monopole of magnetic charge/monopole number $(m,n)$; the $HT$ twist of the resulting theory is topological and we describe a boundary VOA whose category of modules models line operators in the twisted theory. The appearance of $M(3,4)$ as a boundary VOA is via the same mechanism used in Section 4.

We note that this example is not quite in the same spirit of [27] in that we do not expect it to be related to an IR supersymmetry enhancement of some UV $\mathcal{N} = 2$ theory. Rather, we view our example as the $\mathbb{Z}_2$ orbifold of the deformation of a chiral multiplet to a massive chiral multiplet. The deformed bulk theory flows in the IR to a topological theory admitting an interesting VOA on its boundary; a massive chiral admits a free fermion on its boundary and $M(3,4)$ is the even subalgebra thereof. We note that the minimal model $M(3,4)$ is not classically free [45], whereas the free fermion VOA at the boundary of an $HT$-twisted massive chiral is classically free. It is unclear what property of the bulk TQFT is lost upon taking this $\mathbb{Z}_2$ quotient that could translate to the loss of classical freeness of the boundary VOA.

We start with a $(\mathcal{N}, \mathcal{D}, D)$ boundary condition; there is no gauge anomaly as the effective Chern-Simons level is given by

$$k_{eff} = \begin{pmatrix} 0 & 2 \\ 2 & 0 \end{pmatrix}. \qquad (A.2)$$

The mixed Chern-Simons term implies the $U(1)_{2,\partial}$ is broken by this boundary condition. The corresponding half-index takes the form

$$\mathbb{I}_{(\mathcal{N},\mathcal{D},D)}(q) = \sum_{\mathfrak{m}_2 \in \mathbb{Z}} \oint \frac{ds_1}{2\pi i s_1} s_1^{2\mathfrak{m}_2} (s_1^{-1} q^{\frac{1}{2}}; q)_\infty = \sum_{n \geq 0} \frac{q^{2n^2}}{(q,q)_{2n}} = q^{\frac{1}{48}} \chi(V_{1,1}^{(3,4)}), \qquad (A.3)$$

where the penultimate equality is realized by using the $q$-binomial theorem. In the same way, we can obtain the characters for the other two $M(3,4)$ modules: introducing a Wilson line of

charge $(0,1)$ gives

$$
\begin{aligned}
\mathbb{I}_{(\mathcal{N},\mathcal{D},D)}(q)[\mathcal{W}_{(0,1)}] &= \sum_{\mathfrak{m}_2 \in \mathbb{Z}} q^{\mathfrak{m}_2} \oint \frac{\mathrm{d}s_1}{2\pi i s_1} s_1^{2\mathfrak{m}_2}(s_1^{-1}q^{\frac{1}{2}};q)_\infty \\
&= \sum_{n\geq 0} \frac{q^{n(2n+1)}}{(q,q)_{2n}} = q^{\frac{1}{48}-\frac{1}{16}} \chi(V_{1,2}^{(3,4)}),
\end{aligned} \tag{A.4}
$$

and introducing a Wilson line of charge $(1,0)$ (together with a background Wilson line for the $R$-symmetry)

$$
\begin{aligned}
\mathbb{I}_{(\mathcal{N},\mathcal{D},D)}(q)[\mathcal{W}_{(1,0)}] &= \sum_{\mathfrak{m}_2 \in \mathbb{Z}} \oint \frac{\mathrm{d}s_1}{2\pi i s_1} s_1^{2\mathfrak{m}_2}(-q^{-\frac{1}{2}}s_1)(s_1^{-1}q^{\frac{1}{2}};q)_\infty \\
&= \sum_{n\geq 0} \frac{q^{2n(n+1)}}{(q,q)_{2n+1}} = q^{\frac{1}{48}-\frac{1}{2}} \chi(V_{1,3}^{(3,4)}).
\end{aligned} \tag{A.5}
$$

## A.1 Boundary vertex algebra in the $HT$ twist

As in Section 4, we can describe the desired boundary vertex algebra by deforming the algebra of boundary local operators in the $HT$ twist of the undeformed theory. We impose $\mathcal{N}=(0,2)$ Neumann boundary conditions on the first vector multiplet and $\mathcal{N}=(0,2)$ Dirichlet boundary conditions on the second of the vector multiplet and on the chiral multiplet. The perturbative local operators in the $HT$ twist of the undeformed theory can be described as follows, *cf.* Section 6 of [32]. First, there is the current $J_2$ (coming from the gauge fields with Dirichlet boundary conditions) together with the boundary fermion $\overline{\lambda}_1$ (coming from the chiral field). These fields have regular OPEs with themselves and with one another. Infinitesimal gauge transformations of these fields can be described by the following differential

$$
QJ_2(z) = 2\partial c_1(z), \qquad Q\overline{\lambda}(z) = -2 :c_1(z)\overline{\lambda}(z): , \tag{A.6}
$$

where $c_1(z)$ is a fermionic generator of cohomological degree 1 with regular OPEs with all the other two generators and itself. The variation of $J_2$ encodes the effective Chern-Simons level. Gauge invariant local operators in this perturbative subalgebra are realized by removing the zero-mode of $c_1$, considering operators of weight zero, *i.e.* operators built from $\partial c_1$ and $J_2$, and then taking cohomology with respect to $Q$. The only perturbative local operator surviving $Q$-cohomology is the trivial local operator 1.

We now turn to the non-perturbative corrections. These are realized by introducing additional boundary monopole operators $V_{(0,\mathfrak{m})}(z)$ having regular OPEs with all other fields and transforming as

$$
QV_{(0,\mathfrak{m})}(z) = 2\mathfrak{m} :c_1(z)V_{(0,\mathfrak{m})}(z): . \tag{A.7}
$$

The full, non-perturbative algebra of local operators is then realized as $Q$-cohomology of chargeless operators built from the perturbative generators $\partial c_1$, $J_2$, $\overline{\lambda}$ and the boundary monopoles $V_{(0,\pm 1)}(z)$. We note that $V_{(0,\pm 1)}(z)$ has regular OPEs with all other generators. As with the $HT$ twist of $\mathcal{T}_{\min}$, there is a particularly important local operator that has regular OPEs with all of the generating fields, and hence all of the boundary vertex algebra:

$$
T = \tfrac{1}{2} :V_{(0,1)}(\partial\overline{\lambda})\overline{\lambda}: . \tag{A.8}
$$

## A.2 Topological deformation

We now deform the theory by introducing the superpotential $\frac{1}{2}\Phi^2 V_{(0,-1)}$, *cf.* Section 4. At the chain level, this induces a nontrivial OPE for the fermion with itself:

$$
\overline{\lambda}(z)\overline{\lambda}(w) \sim \frac{V_{(0,-1)}(w)}{z-w} . \tag{A.9}
$$

From this OPE, it is a simple computation to verify that the previously-central operator $T$ has the following OPE with $\overline{\lambda}$:

$$T(z)\overline{\lambda}(w) \sim \frac{\frac{1}{2}\overline{\lambda}(w)}{(z-w)^2} + \frac{\partial\overline{\lambda}(w)}{z-w}. \tag{A.10}$$

Moreover, the OPE of $T(z)$ with itself is that of a stress tensor at central charge $\frac{1}{2}$:

$$T(z)T(w) \sim \frac{\frac{1}{4}}{(z-w)^4} + \frac{2T(w)}{(z-w)^2} + \frac{\partial T(w)}{z-w}. \tag{A.11}$$

Unfortunately, the operator $T(z)$ does not quite survive $Q$-cohomology. For example, we find[10]

$$Q : (\partial\overline{\lambda})\overline{\lambda} := -2 : c_1(\partial\overline{\lambda})\overline{\lambda} : - : \partial^2 c_1 V_{(0,-1)} :, \tag{A.12}$$

from which it follows that

$$QT = T - \partial^2 c_1. \tag{A.13}$$

Thankfully, this is an easy problem to fix: we instead consider $\widetilde{T} := T + \frac{1}{2}\partial J_2$. As $J_2$ has regular OPEs with all of the other generators, it follows that we still have

$$\widetilde{T}(z)\overline{\lambda}(w) \sim \frac{\frac{1}{2}\overline{\lambda}(w)}{(z-w)^2} + \frac{\partial\overline{\lambda}(w)}{z-w}, \tag{A.14}$$

as well as

$$\widetilde{T}(z)\widetilde{T}(w) \sim \frac{\frac{1}{4}}{(z-w)^4} + \frac{2\widetilde{T}(w)}{(z-w)^2} + \frac{\partial\widetilde{T}(w)}{z-w}, \tag{A.15}$$

as desired. Based on our index computations, we expect that $\widetilde{T}$ generates the $Q$-cohomology, thereby realizing the minimal $M(3,4)$.

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
