# Peer review of "Boundary vertex algebras for 3d $\mathcal{N}=4$ rank-0 SCFTs"

_SciPost Physics, doi:SciPost Phys. 17, 057 (2024)_

## Round 2 · Referee Report · Anonymous (Referee 1) · 2024-5-16

Report

This paper studies the boundary VOAs of a special family of topologically twisted 3d SCFTs. This is very much a topic of current interest within the hep-th community and a natural continuation of previous work by the authors.

The relevant class of 3d SCFTs is a new and interesting family of Abelian Chern-Simons matter theories, of so-called rank-0 due to the Coulomb and Higgs branches being 0-dimensional, and which have a UV $\mathcal{N}=2$ Lagrangian description that is enhanced to $\mathcal{N}=4$ in the IR regime. The boundary VOA is studied for right Dirichlet boundary conditions and the topological B-twist, which is obtained as a deformation of the HT-twist. In the example of the minimal rank-zero theory, the authors identify in this way the boundary VOA with an affine osp(1|2) current algebra, and discover a new level-rank duality with the Virasoro minimal model M(2,5).

This paper has many strong points; it is well written, with a clear logical structure, and interesting results which are well explained within the wider context of research on VOAs in higher-dimensional SQFTs. One aspect that the authors might consider, and which could improve the clarity of the exposition, would be to add an expanded explanation in Section 4.1 of the OPEs.

There are a few typos to correct: - in equation (2.6), the summation limit should be “r”; - in equation (3.11), $Q_{-,z}$ should be $G_{-,z}$; - in equation (4.12), $\delta_{n+m,0}$ should be $\delta_{n,0}$ for consistency with (4.11) and (4.14).

Recommendation

Publish (surpasses expectations and criteria for this Journal; among top 10%)

  • validity: top
  • significance: top
  • originality: top
  • clarity: high
  • formatting: excellent
  • grammar: excellent

Author:  Heeyeon Kim  on 2024-06-28  [id 4590]

(in reply to Report 1 on 2024-05-16)

We would like to thank the referee for the reports and useful comments. Please see the list of changes (from 1 to 4) we made in the updated version.

---

## Round 2 · Referee Report · Anonymous (Referee 2) · 2024-6-5

Report

This paper investigates boundary vertex operator algebras (VOAs) associated with 3D rank-0 N=4 superconformal field theories (SCFTs). This unique class of SCFTs has drawn much attention recently, as they are the simplest 3D CFTs with 8 supercharges and have no counterparts in higher dimensions. Previous studies suggest that 3D rank-0 SCFTs become finite and semi-simple non-unitary topological quantum field theories (TQFTs) after topological twisting, and support rational chiral algebras at their boundaries.

In this paper, the authors clarify the precise relationship between the rank-0 condition of bulk SCFTs and the rationality of the boundary VOAs. They then explicitly construct rational boundary VOAs for specific examples using recently developed tools. While previous research on rank-0 SCFTs has mainly focused on modular data, this paper is the first to study the underlying VOAs, which are more fundamental. Additionally, the authors explore the intriguing relationship between the boundary VOAs resulting from A- and B-twistings of a single rank-0 SCFT.

This paper is well and carefully written, and easily meets the criteria for publication in SciPost. I strongly recommend its acceptance.

Recommendation

Publish (surpasses expectations and criteria for this Journal; among top 10%)

  • validity: high
  • significance: top
  • originality: high
  • clarity: high
  • formatting: excellent
  • grammar: excellent

Author:  Heeyeon Kim  on 2024-06-28  [id 4591]

(in reply to Report 2 on 2024-06-05)

We would like to thank the referee for the report.

---

## Round 2 · Referee Report · Anonymous (Referee 3) · 2024-6-19

Report

The paper under review studies the class of 3d N=4 SCFTs of "rank zero," meaning those that have a zero-dimensional moduli space (both the Higgs and the Coulomb branch). Typically, such SCFTs do not admit a straightforward Lagrangian description with an obvious N=4 supersymmetry; they tend to appear either as the infrared limits of flows of N=2 theories, or by gauging an infrared emergent symmetry of an N=4 Lagrangian theory. As such, the topological A and B twists of such a theory are not straightforward to compute in a Lagrangian description.

The paper uses the holomorphic-topological twist of such a flow of N=2 theories as a means to arrive at a concrete description of the topological twists of the IR SCFT. Since RG flow is rendered exact in the minimally twisted theory, one already has a concrete description at this level. It is then possible to try and identify the action of the residual supersymmetries on the minimally twisted theory, using these to get a concrete description of the further twist. Such a two-step twisting procedure is also important for understanding holomorphic boundary conditions, which do not arise from supersymmetric boundary conditions that are naively compatible with the topological supercharge.

The paper is primarily interested with studying the observables (VOAs) of holomorphic boundary theories, which has been a topic of great interest of late. In section 2, classes of flows of N=2 theories that define IR N=4 SCFTs with zero-dimensional moduli spaces are reviewed. Section 3 reviews some ideas on the holomorphic-topological twist of an N=2 theory, on the algebra of residual supersymmetries that define further deformations to the A and B twists, and on compatibility of boundary conditions in the holomorphic theory with these deformations. Section 4 identifies an action of residual supersymmetries in the holomorphic twist of the specific N=2 flow T_1, and discusses the VOA supported on a natural (right) Dirichlet boundary condition there and its deformation to the B-twist. The resulting VOA is studied in detail, and another (left) Neumann boundary condition is studied similarly. The two are conjectured to be related by a form of Koszul duality, giving rise to a braid-reversing equivalence between their categories of modules. This arises (speaking roughly) either from an identification of the two as transverse generating objects in some appropriate fashion, or from an identification of both module categories with the category of bulk line operators. Section 5 discusses similar structures for "higher-rank" families T_r of N=2 flows, at a somewhat more speculative level.

I enjoyed reading this paper, and it clearly makes a contribution to an important and active area. The central results have to do with the theory T_1, and these are worked out cleanly and in detail. I am happy to recommend publication of the paper in SciPost, but will allow myself to detail some comments and suggestions here, which the authors can implement as they see fit. In my opinion, taking a few of these minor suggestions into account would increase the readability of the paper without much additional effort.

General remarks:
The notion of "rank" seems to appear twice in incompatible fashion. In the title of the paper, it means the dimension of the SCFT moduli space, at least implicitly. In section 2.2, the theories T_r are defined as "a class of rank-zero theories." But in section 5, we refer to the "higher-rank theory T_r," here referring to the rank of the gauge group in the underlying N=2 Lagrangian. Perhaps this clash is unavoidable, but it could be made a bit clearer that it is happening... of course, it would be ideal to refer to the "dimension of the moduli space" for the first of the two, if possible.

p. 2: "expected to extend." This confuses me slightly; shouldn't an action of the *local* Lie algebra of holomorphic vector fields (of appropriate type) imply an action of the Virasoro algebra on local operators, by placing the theory on a specific geometry, such as C with one marked point?

Also, a brief and clear reminder about the definitions of the terms being used ("lisse," "C2-cofinite") and the implications between them would improve the paragraph above "Q:", at least for non-expert readers.

p. 3, point (i): "encoded into." Based on the conjecture described in the next sentence, shouldn't this mean something more concrete, like "isomorphic to" or "identified with"?

p. 3, last sentence: Maybe emphasize conditions 1 and 2 typographically a bit more. Since these appear hidden in the body text above, it is easy to mistakenly look at the numbered points (i) and (ii) instead; in turn, these don't seem to be referred to by number again later. So the choice of typographical emphasis feels a bit misleading.

p. 7: The way the R-symmetry is referred to at the top of the page isn't consistent with how the diagonal torus is described in the third paragraph. Do the authors just want to use the global form Spin(4) throughout? Since spinorial representations of R-symmetry can appear in the multiplets of 3d N=4 SUSY, this seems potentially appropriate, and avoids needing to emphasize the Z/2 quotient.

p. 9: The last paragraph above section 3.3 appears unfinished; it discusses a regrading of the theory, but not an actual deformation of the differential.

p. 10: Consider defining the notation (\mathcal{D},D) explicitly here. This is done implicitly, but the notation recurs later as well.

p. 16: "agree with one another up to a factor of -1 [...] and hence are linearly dependent." The second part is clear, and can be safely omitted.

p. 17, footnote 5: Is it worth commenting a bit more on why this mismatch occurs, or what the significance of the fact that the GKSLY matrices do not satisfy these relations should be? An explanatory sentence would help the reader here.

p. 20: "deformation" in last sentence of first paragraph of 5.1.1 should be "deform." Also, why are higher L-infinity operations a problem, or undesirable? They may or may not be there, but this is a consequence of the form of the superpotential, and so just a feature of the theory...

p. 21: "generic/deformed Dirichlet boundary conditions" are defined for the first time here, but referred to earlier. Perhaps just define this term where it first appears.

Recommendation

Publish (easily meets expectations and criteria for this Journal; among top 50%)

  • validity: -
  • significance: -
  • originality: -
  • clarity: -
  • formatting: -
  • grammar: -

Author:  Heeyeon Kim  on 2024-06-28  [id 4592]

(in reply to Report 3 on 2024-06-19)

We would like to thank the referee for the report and useful comments. Please see the list of changes we made (from 5 to 17) in the updated version.

---

## Round 3 · Referee Report · Anonymous (Referee 1) · 2024-7-1

Report

The authors thoughtfully addressed the comments from my previous report on this paper. I was already in favour of publishing the first version sent to me, independently of the edits that I suggested. Now I recommend publication without any further comments.

Recommendation

Publish (surpasses expectations and criteria for this Journal; among top 10%)

---

## Round 3 · Referee Report · Dongmin Gang (Referee 4) · 2024-7-2

Report

The manuscript has been improved and is now suitable for publication without further modifications.

Recommendation

Publish (surpasses expectations and criteria for this Journal; among top 10%)

---

## Round 3 · Referee Report · Anonymous (Referee 3) · 2024-7-10

Report

I thank the authors for their careful response to the suggestions in the previous reports. The edits have improved the paper and the exposition; I have no further suggestions, and am happy to recommend publication in SciPost Physics in this form.

Recommendation

Publish (easily meets expectations and criteria for this Journal; among top 50%)

---

## Round 3 · List of Changes

1. Elaborated on the OPEs of the various generators appearing in Section 4.1.

1) expanded on OPEs of perturbative generators in the $HT$ twist below Eq. (4.10) 2) spelled out OPEs of perturbative generators with boundary monopoles in the $HT$ twist below Eq. (4.14); added OPEs of boundary monopoles 3) provided example of how deformed OPEs can be deduced from associativity

  1. Changed $n$ to $r$ in Eq. (2.6).

3.Changed $Q_{-,z}$ to $G_{-,z}$ in Eq. (3.11)

4.Changed $\delta_{n+\mathfrak{m},0}$ to $\delta_{n,0}$ in Eq. (4.12)

  1. Changed rank $\to$ level for describing the theories $\mathcal{T}_r$

  2. p.2, reworded sentence to simply say the operator realizes an action of the Virasoro algebra.

  3. Added comment that $C_2$-cofiniteness is equivalent to $R_\mathcal{V}$ being finite-dimensional and that a vertex algebra is lisse if its singular support (as a module for itself) is 0-dimensional.

  4. p.3, changed encoded into '' toidentified with''

  5. p.3, placed conditions 1) and 2) in an enumerate environment.

  6. p.7, changed $SO(4)_R$ to Spin$(4)_R$

  7. p.9, added a sentence at the end of section 3.2 about the deformation of the differential.

  8. p.10, added an explicit statement of the boundary conditions $\mathcal{D}$ and $D$.

  9. p. 16, removed the explicit statement of linear dependence.

  10. p. 17, added a comment saying that the $T$-matrix of GKLSY is only determined up to an overall phase.

  11. p.20, changed deformation'' todeform''

  12. Added a footnote describing why we want a boundary condition without any higher operators.

  13. Added a definition of the generic Dirichlet boundary condition $D_c$ when it firsts appears in the paragraph before Section 4.1.

---

## Editorial Decision

published